# Sensitivity of deep ocean biases to horizontal resolution in prototype CMIP6 simulations with AWI-CM1.0

Thomas Rackow[1], Dmitry Sein[1,2], Tido Semmler[1], Sergey Danilov[1], Nikolay Koldunov[1,3], Dmitry Sidorenko[1], Qiang Wang[1], and Thomas Jung[1,4]

[1]Alfred Wegener Institute, Helmholtz Centre for Polar and Marine Research, Bremerhaven, Germany
[2]Shirshov Institute of Oceanology, Russian Academy of Science, Moscow, Russia
[3]MARUM, University of Bremen, Bremen, Germany
[4]Institute of Environmental Physics, University of Bremen, Bremen, Germany

*Correspondence to:* Thomas Rackow (thomas.rackow@awi.de)

**Abstract.** CMIP5 models show substantial biases in the deep ocean that are larger than the level of natural variability and the response to enhanced greenhouse gas concentrations. Here we analyse the influence of horizontal resolution in a hierarchy of five multi-resolution simulations with the AWI Climate Model (AWI-CM), which employs a sea ice-ocean model component formulated on unstructured meshes. The ocean grid sizes considered range from a nominal resolution of $\sim 1°$ (CMIP5-type) up to locally eddy-resolving. We show that increasing ocean resolution locally to resolve ocean eddies leads to reductions in deep ocean biases, although these improvements are not strictly monotonic for the five different ocean grids. A detailed diagnosis of the simulations allows to identify the origins of the biases. We find that two key regions at the surface are responsible for the development of the deep bias in the Atlantic Ocean, the north-eastern North Atlantic and the region adjacent to the Strait of Gibraltar. Furthermore, the Southern Ocean density structure is equally improved with locally explicitly resolved eddies compared to parameterized eddies. Part of the bias reduction can be traced back towards improved surface biases over outcropping regions, which are in contact with deeper ocean layers along isopycnal surfaces. Our prototype simulations provide guidance for the optimal choice of ocean grids for AWI-CM to be used in the final runs for phase 6 of the 'Coupled Model Intercomparison Project' (CMIP6) and for the related flagship simulations in the 'High Resolution Model Intercomparison Project' (HighResMIP). Quite remarkably, retaining resolution only in areas of high eddy activity along with excellent scalability characteristics of the unstructured-mesh sea ice-ocean model enables us to perform the multi-centennial climate simulations needed in a CMIP context at (locally) eddy-resolving resolution with a throughput of 5–6 simulated years per day.

## 1 Introduction

Biases at the ocean surface are relatively well studied (e.g. Wang et al., 2014a). However, climate models also suffer from less known biases in the deep ocean that have the potential to impact the storage of heat by the ocean. This issue may be of relevance for projections of the future climate performed in the framework of the Coupled Model Intercomparison Project (CMIP; Taylor et al., 2012).

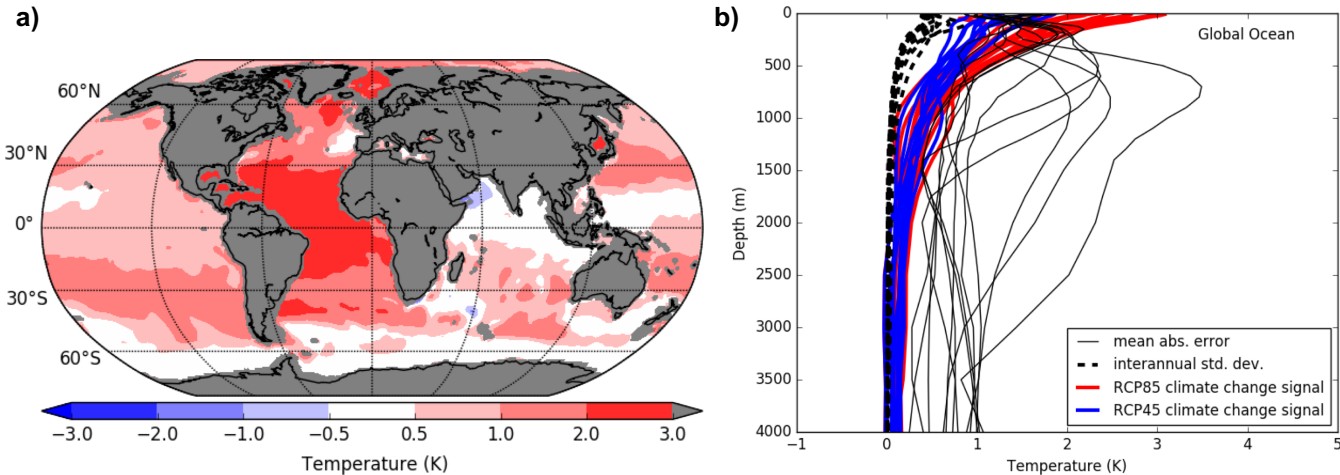

**Figure 1.** Biases in CMIP5 models with respect to the PHC climatology (PHC 3.0, updated from: Steele et al., 2001). **a)** Ensemble mean DJF potential temperature bias [K] at a depth of 1000 m in 13 CMIP5 historical simulations for the period 1971—2000. **b)** Individual depth-profiles of the mean absolute potential temperature error in the considered CMIP5 models (black lines). The interannual standard deviation [K] (black dashed lines) and the RCP4.5 and RCP8.5 climate change signal [K] (2071–2100 minus 1971–2000; blue and red lines) are given for comparison.

A major bias present in CMIP5 models is reflected by a too warm and saline deep ocean compared to observations (e.g. in the EC-Earth model; Sterl et al., 2012). This systematic error (Table 1) is illustrated by comparing temperature profiles from 13 CMIP5 historical runs (Fig. 1b) with the PHC3 climatology (PHC 3.0, updated from: Steele et al., 2001). Importantly, the mean absolute error in deeper ocean layers is larger than the interannual variability (the standard deviation of annual means). It is also

larger than the climate change signal as determined from RCP8.5 and RCP4.5 emission scenarios. Or formulated differently, deep ocean biases are larger than the signals we aim to predict, which may be cause for concern in non-linear systems. When considering horizontal maps of the multi model-mean potential temperature bias in 1000 m depth (Fig. 1a), one can clearly see that the largest bias is located in the Atlantic sector. As seasonal variability is low in 1000 m depth, the bias is very similar for different seasons (not shown). Although one could argue that this error is "well-hidden" from the atmosphere, thus having

little impact on atmospheric parameters, it has the potential to change the outcropping region and position of isopycnals. This could lead to a wrong "mapping" of the deep ocean to the surface; in other words, this could link the deep ocean to incorrect locations at the surface, which may result in erroneous water mass formation. In turn, this can potentially have significant effects on the heat uptake of the deep ocean, thus impacting climate change projections. As an example, the magnitude of the projected climate change in the ocean appears to be ordered according to the models' mean absolute errors (Table 1).

Previous work has identified an important role for mesoscale eddies, showing that they act "as a barrier or gatekeeper to heat penetration from the surface into the ocean interior" (Hewitt et al., 2017) by counter-acting the downward heat transport from the mean ocean circulation (Griffies et al., 2015; von Storch et al., 2016). With the increase in simulated eddy activity when increasing resolution towards 0.1°, the magnitude of vertical eddy heat transport also increases, which in turn reduces

temperature drifts in the simulated deep ocean when compared to coarse-resolution ocean models of about $1°$ (Hewitt et al., 2017; Griffies et al., 2015). The physical mechanism behind this upward eddy heat transport is the mixing of heat along inclined surfaces of constant density (isopycnals) by eddies and eddy-induced transport. However, the position and tilt of the isopycnals themselves is also strongly impacted by mesoscale eddies, which can influence the mapping from the surface ocean layers to the deeper ocean. Since globally eddy-resolving climate simulations are still very expensive in a CMIP-context, and since current eddy parameterizations do not seem to capture vertical eddy fluxes to full degree (Hewitt et al., 2017), local refinement to explicitly resolve regions of high eddy activity is thus a promising approach to tackle deep-ocean biases (Zadra et al., 2017).

To study the impact of horizontal resolution on the biases in the deep ocean, the AWI Climate Model (AWI-CM; Sidorenko et al., 2015; Rackow et al., 2016) is employed in this work. The 'deep bias' can be reproduced in the AWI-CM 'benchmark' configuration that has a rather coarse nominal ocean resolution of $\sim 1°$ typically employed in CMIP5 (not shown). Therefore, the model is well-suited to study the impact that locally enhanced resolution can have on deep ocean biases in CMIP5 models. In order to test the hypothesis that locally too coarse spatial resolution is responsible for the development of the deep ocean biases, we gradually increase the number of ocean grid points in four additional AWI-CM configurations with otherwise identical settings and parameter choices. It is shown that the strong 'deep bias' in the North Atlantic reduces with higher resolution to rather small values that are comparable to those found in other ocean basins. Together with a competitive throughput of 5–6 simulated years per day for the highest analyzed resolutions, this gives a strong case to aim for a high resolution (10 km and higher) in eddy-active regions not only in HighResMIP (Haarsma et al., 2016), but already for AWI's CMIP6 standard configuration.

The paper is structured as follows. Section 2 introduces the model configurations and the hierarchy of ocean meshes with systematically increasing spatial resolution in the North Atlantic. The sensitivity of vertical profiles and horizontal maps of surface and interior biases to increasing spatial resolution is studied in section 3, as well as the development of deep ocean biases along relevant surfaces of constant density. The paper closes with a conclusion and further discussions in section 4.

## 2 Model configuration

The AWI-CM (formerly ECHAM6-FESOM; Sidorenko et al., 2015; Rackow et al., 2016) is a coupled configuration in which ECHAM 6.3.01 (Stevens et al., 2013) is coupled to the Finite Element Sea Ice-Ocean Model (FESOM1.4; Wang et al., 2008; Timmermann et al., 2009; Sidorenko et al., 2011; Wang et al., 2014b). It supports unstructured multi-resolution grids for the ocean and sea ice and has shown good performance in simulating present-day climate when compared to more traditional regular-grid climate models participating in CMIP5 in terms of both the mean climate state (Sidorenko et al., 2015) and climate variability (Rackow et al., 2016). Compared to the coupling procedure detailed in the above mentioned studies, the model now uses a bicubic mapping for the interpolation of the wind-stress components to the ocean grid in order to better conserve higher-order properties like the curl (Valcke, 2013), using OASIS3-MCT (Craig et al., 2017). In this study, we will analyze monthly-mean output of five 100yr-long pre-industrial simulations. The simulations are initialized from the PHC climatology (PHC 3.0, updated from: Steele et al., 2001) and zero velocities. The ocean model does not apply geothermal

**Table 1.** CMIP5 models considered in the illustration of the deep ocean bias in Fig. 1, in decreasing order according to their mean absolute potential temperature error at 1000 m depth. The absolute error is computed at every gridpoint as the absolute difference $|T_m - T_o|$, where $T_o$ is the observed and $T_m$ is the modeled potential temperature.

| CMIP5 model | mean absolute error for global ocean [K] | interannual std. dev. for global ocean [K] | climate change signal RCP4.5 for global ocean [K] | climate change signal RCP8.5 for global ocean [K] |
|---|---|---|---|---|
| GISS-E2-R | 3.11 | 0.08 | 0.61 | 0.73 |
| MPI-ESM-LR | 2.43 | 0.12 | 0.38 | 0.47 |
| GFDL-CM3 | 2.02 | 0.06 | 0.44 | 0.51 |
| ACCESS1-3 | 1.94 | 0.09 | 0.48 | 0.59 |
| IPSL-CM5B-LR | 1.35 | 0.05 | 0.40 | 0.43 |
| GISS-E2-H | 1.03 | 0.06 | 0.41 | 0.52 |
| CCSM4 | 0.87 | 0.05 | 0.40 | 0.48 |
| HadGEM2-ES | 0.87 | 0.07 | - | 0.31 |
| NorESM1-ME | 0.74 | 0.05 | 0.39 | 0.51 |
| CMCC-CM | 0.68 | 0.04 | 0.26 | 0.33 |
| CanESM2 | 0.66 | 0.04 | 0.37 | 0.46 |
| MRI-ESM1 | 0.55 | 0.04 | - | 0.26 |
| MRI-CGCM3 | 0.54 | 0.04 | 0.16 | 0.22 |

This is based on the DJF season and historical runs for the period 1971–2000; for RCP4.5 and RCP8.5 the climate change signal is based on the period 2071-2100 compared to the historical period 1971-2000

heating as lower boundary condition (e.g., Adcroft et al., 2001; Downes et al., 2016). In order to parameterize eddies at non-eddy resolving resolutions, the Gent and McWilliams (1990) parameterization (GM) is applied with isoneutral diffusion (Redi, 1982). All prototype simulations use a reference diffusivity $K_{\mathrm{ref}}(x, y) = 600 \, \mathrm{m^2 \, s^{-1}}$, which is scaled by the local resolution (Wang et al., 2014b), and a GM coefficient $K_{\mathrm{GM}} = K_{\mathrm{ref}}/2$. As detailed by Wang et al. (2014b), tapering functions following
5 Danabasoglu and Mc Williams (1995) and Large et al. (1997) are also applied to $K_{\mathrm{GM}}$. Depending on the local resolution, the GM parameterization in FESOM1.4 is smoothly switched off at resolutions smaller than 25 km (red areas in Fig.2), and its effect increases linearly until 50 km, when the parameterization is fully active (Wang et al., 2014b). For example, the parameterization is locally switched off when using the 'MR' and 'HR' meshes, which are locally eddy-resolving, and it is generally active in the lower-resolution 'LR' mesh (see next sections). At mid-latitudes, the Rossby radius is between 25 and 50 km, which is why
10 this simple choice was made. Still, the thresholds of 25 km and 50 km can be considered to be tuning parameters and were chosen in stand-alone simulations with FESOM1.4 using the LR grid. For the Arctic, changing the thresholds can result in too diffuse boundary currents (Wang et al., 2014b). Ultimately, these thresholds should be chosen automatically and separately for differently resolved regions of the global ocean. Their optimal choice thus remains an important research topic for multi-resolution climate applications.

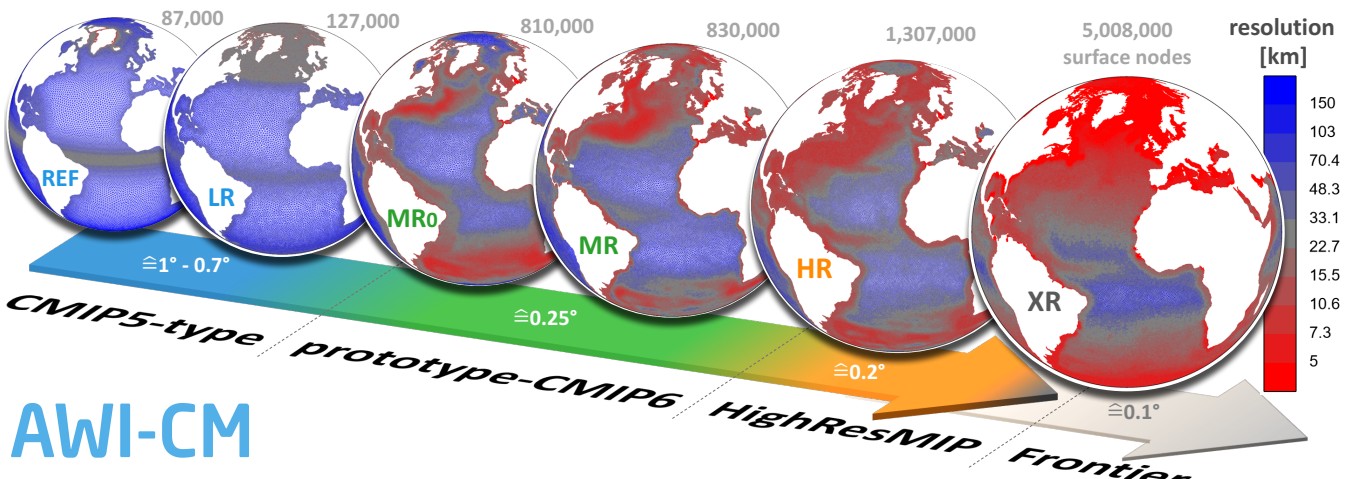

**Figure 2.** Hierarchy of a set of different ocean grid resolutions that are used in this study. The number of surface grid points increases from left to right and, specifically, the spatial resolution in the North Atlantic Ocean increases from REF up to XR. REF and LR use CMIP5-type spatial resolution, with moderate refinement to about 25km in the tropics and in the Arctic (Rackow et al., 2016; Sidorenko et al., 2011). MR0, MR, and HR are medium- and high-resolution meshes, following a different mesh design strategy (Sein et al., 2016), and focus on the Agulhas and North Atlantic current region. The resolution of the frontier mesh (XR) additionally follows the local Rossby radius of deformation, and is capped at 4 km (7 km) in the Arctic (Antarctic) (Sein et al., 2017). White numbers indicate the approximate spatial resolution of corresponding quasi-Mercator grids with the same number of (wet) surface nodes. The GM parameterization is switched off within the red areas ($\leq 25$ km).

**Table 2.** Model settings for the different AWI-CM configurations

| AWI-CM configuration | (previous) ocean mesh name | 2D ocean grid points | atm. resolution | time step FESOM | time step ECHAM6 | coupling | CPU cores (FESOM+ECHAM) | sim. years per day (SYPD) |
|---|---|---|---|---|---|---|---|---|
| REF | ref87k | 86,803 | T63 | 30 min | 450 s | 1h | 384+192 | 21.8 |
| LR* | core2 | 126,859 | T127 | 15 min | 200 s | 1h | 192+576 | 5.6 |
| MR0 | aguv | 810,471 | T127 | 7.5 min | 200 s | 1h | 2304+1152 | 6.2 |
| MR* | glob | 830,305 | T127 | 10 min | 200 s | 1h | 1920+1152 | 6.4 |
| HR* | bold | 1,306,775 | T127 | 10 min | 200 s | 1h | 2400+1200 | 5.5 |
| XR (ocean-only) | fron | 5,007,727 | - | 4 min | - | - | 7200 | 1.5–2 |

*more details on the AWI-CM CMIP6 configurations at https://github.com/WCRP-CMIP/CMIP6_CVs/blob/master/CMIP6_source_id.json (as of June 2019)

## 2.1 Target resolution

In order to find an optimal mesh for the CMIP6 configuration and the associated endorsed Model Intercomparison Projects (MIPs), we performed a hierarchy of prototype pre-industrial CMIP6 simulations with AWI-CM, run at different ocean resolutions (Table 2). Ultimately, we will target coupled configurations with a globally eddy-resolving mesh, which implies "resolving the Rossby radius" almost everywhere with at least 2 grid intervals per Rossby radius (Hallberg, 2013). Using this criterion, we have recently reported on the development of such a 'frontier' mesh (XR; see Fig. 2, right globe), with resolution capped at 4 km (7 km) in the Arctic (Antarctic) (Sein et al., 2017). Sein et al. (2017) note that an even finer resolution will be required locally to fully capture mesoscale eddies. A lot of engineering goes into the creation of such meshes, balancing computational resources and simulation quality, since the multi-resolution approach allows for a flexible distribution of the grid points. It is not clear a priori how best to distribute a fixed number of degrees of freedom over the globe, and Sein et al. (2016) have coined the term "mesh design" for this non-trivial task. The XR mesh has about 5 million surface nodes, which is roughly comparable to a 1/10° quasi-Mercator mesh with about 5–6 million (wet) nodes. However, as of today, the XR mesh is still too computationally demanding for the multi-centennial simulations needed in a CMIP context. Therefore, here the idea is to retain some of the beneficial properties of the XR ocean-only simulation analyzed by Sein et al. (2017) by keeping higher resolution only in hotspots of high eddy activity. This reduces the computational cost to a level that is suitable for multi-centennial coupled climate simulations and ensemble simulations.

## 2.2 Hierarchy of ocean meshes

The hierarchy of different ocean grid resolutions that are used in this study is shown in Fig. 2. The number of surface grid points increases from left to right and, specifically, the spatial resolution in the North Atlantic Ocean systematically increases from 'REF' (reference or 'benchmark' mesh) up to 'HR' (high-resolution). In order to isolate the impact of horizontal resolution, the vertical levels were left unchanged: there are in total 46 levels with vertical resolution ranging from 10 m at the surface to 250 m below 2150 m. Certainly, going to even higher resolutions beyond the XR mesh, a higher number with different placement of levels might need to be considered, but we kept the standard levels in all meshes for consistency. The bathymetry in the different grids is based on a blend of the IBCAO (Jakobsson et al., 2008) and GEBCO (2008) bottom topography data sets, as detailed by Wang et al. (2014b).

'REF' and 'LR' (low-resolution) use CMIP5-type spatial resolution ($\sim 1°$–$0.7°$) with moderate isotropic refinement to about 25 km in the tropics and in the Arctic. The LR mesh was used for ocean-only simulations within the CORE-II intercomparison project (Danabasoglu et al., 2014; Wang et al., 2016a, b) while REF was used as a 'benchmark' mesh for the coupled AWI-CM (Rackow et al., 2016). Although all prototype simulations except REF use a T127 atmosphere (Table 2), we still include the REF/T63 benchmark configuration here for better comparability with previous studies (Sidorenko et al., 2015; Rackow et al., 2016).

The medium-resolution 'MR0' and 'MR' meshes as well as the high-resolution 'HR' mesh follow the new mesh design strategy introduced by Sein et al. (2016). The main approach is to increase resolution locally over areas of high *observed* eddy

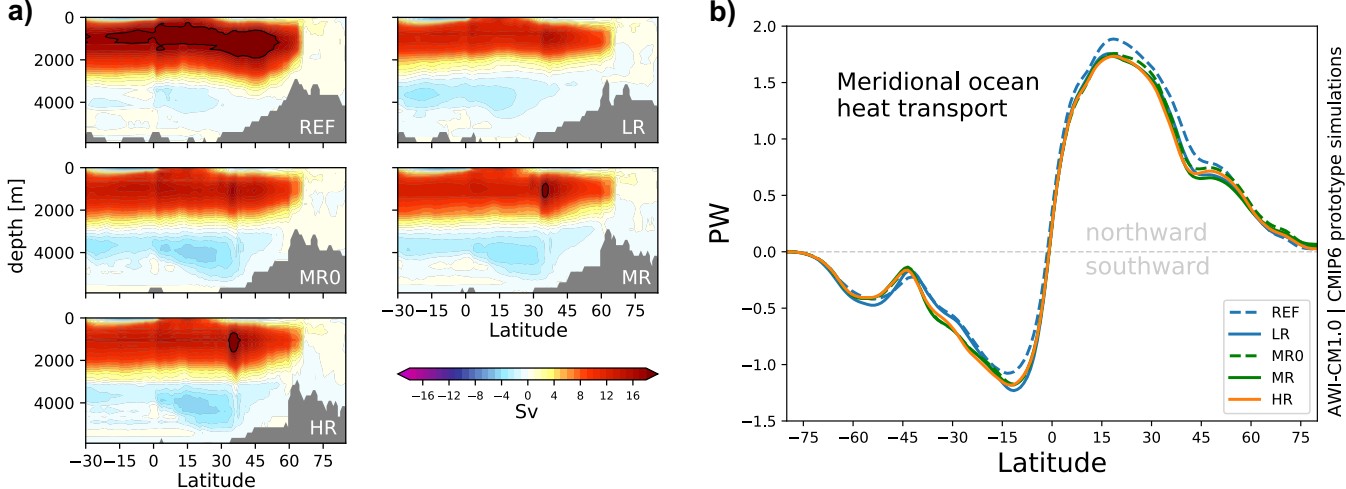

**Figure 3. a)** Atlantic Meridional Overturning Circulation (AMOC) streamfunction [1 Sv = $10^6$ m$^3$/s] and **b)** meridional ocean heat transport [1 PW = $10^{15}$ W] in the five pre-industrial experiments (years 71–100). The black contour in a) denotes the 18 Sv streamline.

variability. While the number of grid points for MR0 and MR is kept at a similar level, MR0 focuses more grid points in the Agulhas region than MR, which in turn focuses them in the North Atlantic Current region (Fig. 2). The HR grid is more balanced in this respect and further increases the size of the areas that use locally increased resolution, resulting in an increase of the number of surface grid points by more than 60%.

It is worth mentioning that HR uses 1.3 million surface grid points (Table 2), similar to traditional 1/4° quasi-Mercator grids (about 1.5 million nodes, of which about 1 million are wet). However, the degrees of freedom on HR are differently distributed, focusing resolution on hotspots of high eddy activity such as the western boundary currents and the Southern Ocean. In fact, this configuration reaches ocean resolutions as high as 1 km locally, e.g. in the Bosporus or over the Danish straits—but still runs at a competitive throughput of 5–6 simulated years per day due to the excellent nearly linear scalability of the FESOM

model (e.g. Biastoch et al., 2018). For a spatial map of the eddy-permitting and eddy-resolving regions on the HR grid, please refer to Fig. 4c in Sein et al. (2016).

## 3   Results

### 3.1   Atlantic Meridional Overturning Circulation (AMOC) and ocean transports

The AMOC pattern is similarly simulated between the model experiments (Fig. 3a). However, it appears that the change of
the atmospheric resolution from REF/T63 to T127 (used in all other configurations) reduces the maximum AMOC strength significantly, which fits to the earlier result by Sein et al. (2018). In contrast, with increasing ocean resolution in the North Atlantic, the AMOC maximum slightly increases to more than 18 Sv in MR and HR (Fig. 3a). The ocean heat transport (Fig. 3b) reflects the behaviour of the AMOC: a stronger poleward ocean heat transport is seen in REF, while LR, MR0, MR, and

**Table 3.** Transport of the ACC at Drake Passage [1 Sv = $10^6$ m$^3$/s]

| AWI-CM configuration | ACC transport |
|---|---|
| REF | 153.7 Sv |
| LR | 213.1 Sv |
| MR0 | 186.1 Sv |
| MR | 186.4 Sv |
| HR | 195.6 Sv |
| CMIP5* | 155 ± 51 Sv |
| Observational estimate** | 136.7 ± 7.8 Sv |

*Meijers et al. (2012), **Cunningham et al. (2003)

HR differ only in details. Concerning the transport of the Antarctic Circumpolar Current (ACC) at Drake Passage (Table 3), REF/T63 somewhat stands out while the other configurations with T127 atmosphere are in closer agreement. REF is close to an observational estimate for the ACC transport, which is $136.7 \pm 7.8$ Sv (Cunningham et al., 2003). The mean ACC transport of current CMIP5 models is however $155 \pm 51$ Sv after Meijers et al. (2012), with range from 90 Sv up to 264 Sv. This means that all analysed configurations simulate an ACC transport within the typical model spread.

We conclude that all five experiments, with vastly different spatial ocean resolution and computational demand, depict a canonical overturning circulation and very similar northward ocean heat transport. These large-scale transport patterns, however, do not necessarily reflect possible differences in the hydrography of the deep ocean. In the following, we will therefore analyse temperature and salinity in more detail.

## 3.2 Vertical profiles of temperature and salinity

Temperature and salinity show major improvements for medium- and high-resolution configurations, as seen from horizontally averaged temperature and salinity profiles for years 71–100 of the pre-industrial simulations (Fig. 4). Differences in the simulated potential temperature and salinity compared to the PHC climatology peak at a depth of around 1000 m. The North Atlantic deep biases, identified both in CMIP5 models (Fig. 1) and in the benchmark REF/T63 and LR/T127 versions of AWI-CM, successively decrease with increasing ocean resolution, both for potential temperature and for salinity (Fig. 4). Although the changes are not strictly monotonic when moving from REF to HR, this highlights the benefit of enhanced spatial resolution. The simultaneous change of the ocean and atmospheric resolution from REF/T63 to LR/T127 leads to a clear improvement of the salinity profiles below 1500 m, and all configurations with T127 atmosphere (LR, MR0, MR, and HR) share a very similar salinity bias in this range. While it is difficult to say what the relative influence is between the atmospheric resolution change (T63 vs T127) and the switch of the ocean grid (REF vs LR), it appears that surface conditions can significantly impact deep

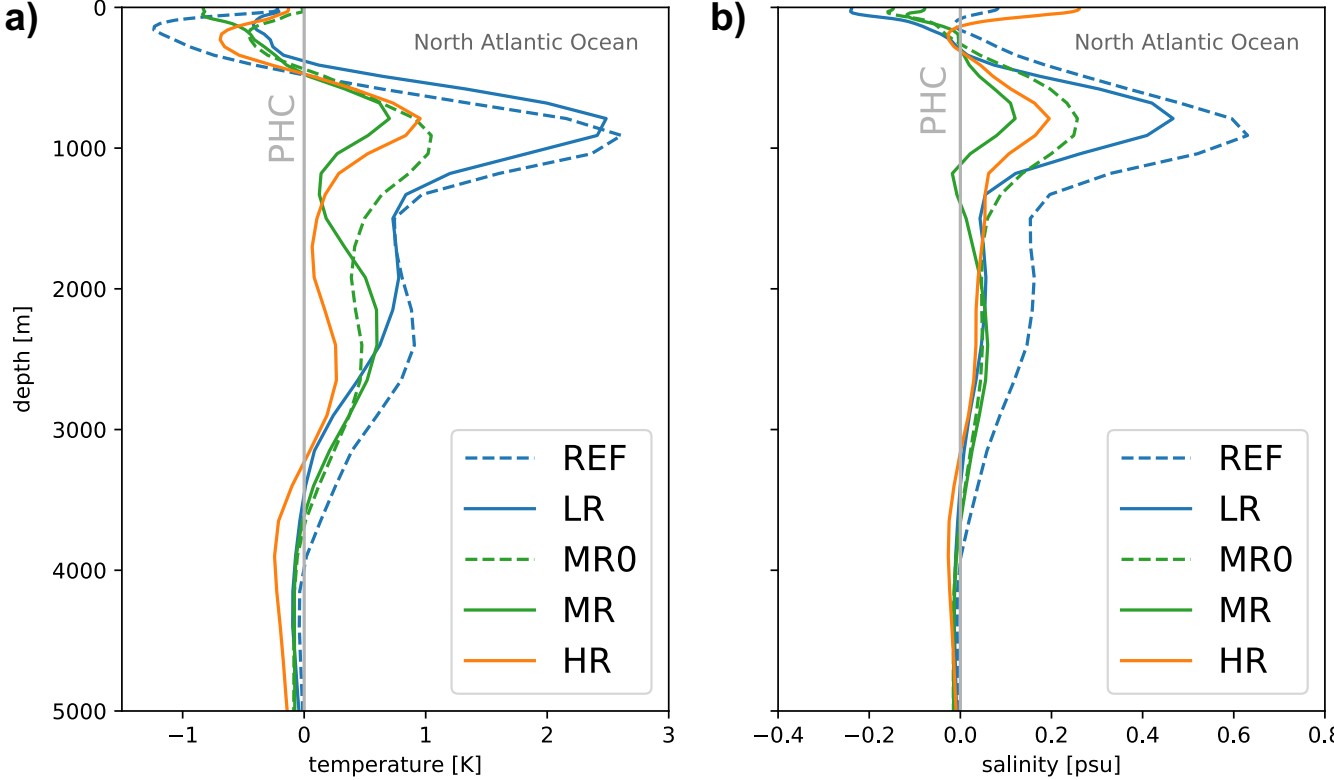

**Figure 4.** Profiles of potential temperature **a)** and salinity **b)** in the North Atlantic Ocean for years 71–100 of the pre-industrial simulations. Shown is the mean difference to the PHC climatology (PHC 3.0, updated from: Steele et al., 2001). With the medium- and high-resolution meshes, the biases around 1000 m depth decrease strongly for both temperature and salinity.

ocean biases. Note that the slight drift in HR towards colder temperatures in the 3000–5000 m range is due to a production of denser waters around Antarctica, coinciding with a stronger deep overturning cell in this model configuration.

### 3.3 Hovmoeller diagrams for temperature and salinity drift

In addition to considering biases at the end of the 100yr-simulations discussed above, it is instructive to study the transient development of the biases over time. To this end, time-depth Hovmoeller diagrams (Griffies et al., 2015; von Storch et al., 2016; Hewitt et al., 2017) have been computed for both potential temperature and salinity. The REF and LR configurations show a strong erroneous initial warming at a depth of around 1000 m together with a cooling in the upper ocean above about 400 m (Fig. 5). In the medium- and high-resolution configurations, both the erroneous deep ocean warming and upper-ocean cooling are reduced. Consistent with the study by von Storch et al. (2016), a similar pattern holds for salinity, with freshening in the upper ocean and salinization in the deep ocean. The improvement of the salinity field with increased spatial resolution is similar to the potential temperature case (Fig. 6).

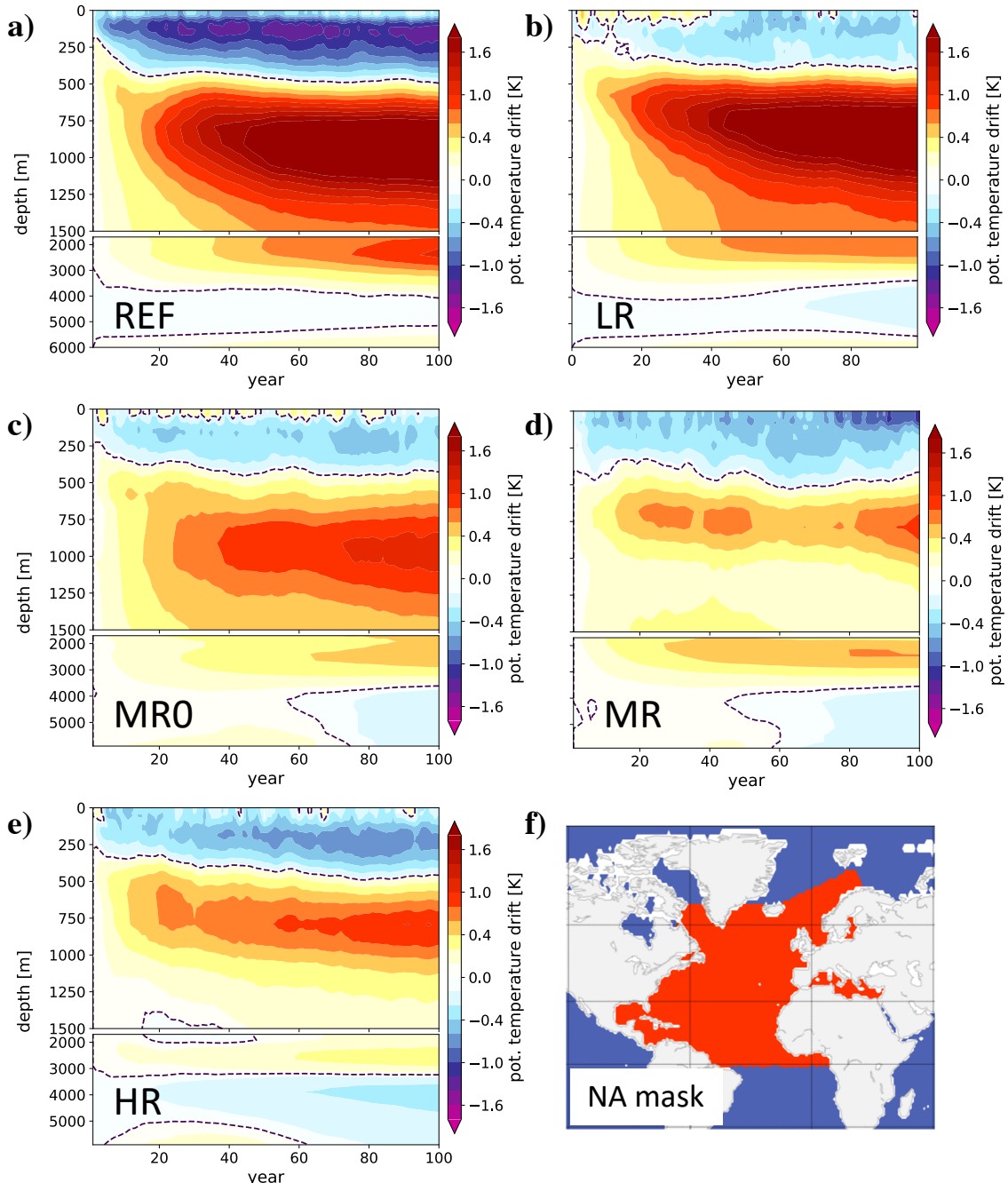

**Figure 5.** Time-depth Hovmoeller diagram of the potential temperature drift [K] in the North Atlantic for **a)–e)** the five pre-industrial simulations. **f)** Definition of the North Atlantic mask that was used in the Hovmoeller analysis.

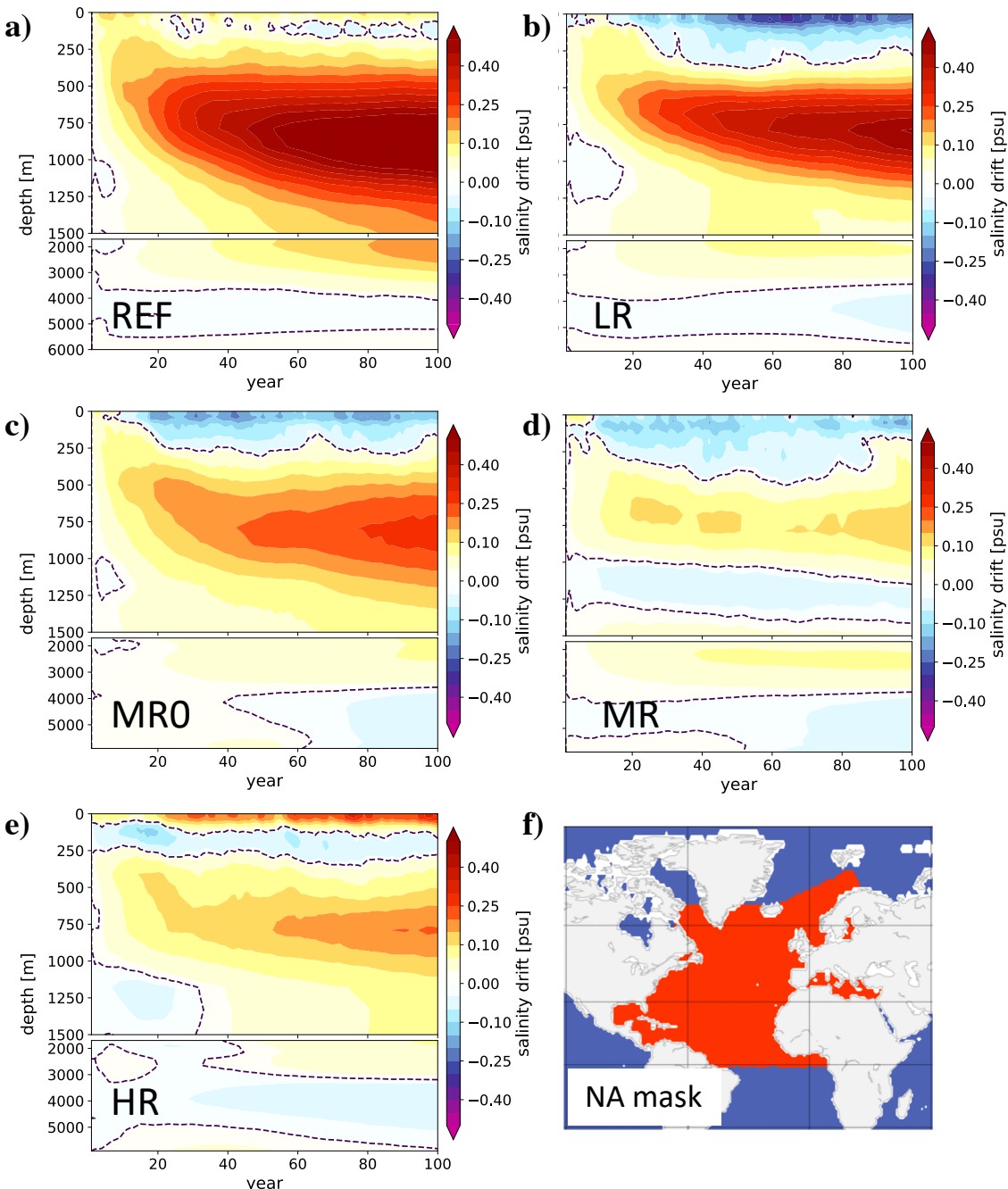

**Figure 6.** Time-depth Hovmoeller diagram of the salinity drift [psu] in the North Atlantic for **a)–e)** the five pre-industrial simulations. The definition of the North Atlantic mask is identical to the one used in Fig.5.

### 3.4 Spatial patterns of temperature and salinity biases

#### 3.4.1 Deep ocean (1000 m)

When considering horizontal maps of potential temperature and salinity biases in the deep ocean, the REF and LR configurations show an erroneous warming in the deep Atlantic ocean (Fig. 7a), similar to the pattern identified for the CMIP5 models (Fig. 1a). With increasing resolution in the North Atlantic, there is a very consistent improvement in deep ocean hydrography (Fig. 7), making the remaining biases in MR and HR comparable in magnitude to smaller biases in the other ocean basins. Compared to the changes in the Atlantic, the other basins remain largely unchanged, suggesting that resolution changes in distant regions play a minor role for the bias reduction in the Atlantic.

It appears as if the resolution increase of MR and HR leads to overshooting close to the Strait of Gibraltar since both MR and HR change the sign of the potential temperature and salinity biases at 1000 m. We hypothesize that at these resolutions, smaller issues become relatively more apparent, that is other processes might need to be included for a proper simulation of the Strait of Gibraltar outflow and spreading of Mediterranean Waters into the North Atlantic. Also, resolving the overflow processes at the Strait of Gibraltar would require resolutions on the order of tens of meters in the horizontal (Izquierdo and Mikolajewicz, 2018) and meters in the vertical direction, which is still far from the resolutions applied in this study. Two possible solutions are therefore the use of an overflow parameterization (e.g., Wu et al., 2007), which is currently not implemented in the model, or systematic changes to the bottom (and lateral) topography at the outflow of the Strait of Gibraltar (Fig. 8a) to minimize spurious entrainment. Vertical profiles of regionally-averaged potential temperature (Fig. 8b) and salinity (Fig. 8c) in the vicinity of the Strait of Gibraltar show that REF/LR (and MR0) generate too much Mediterranean Outflow waters at 1000 m depth, while MR and HR lack these at a depth of 1000 m. Since the simulated model profiles envelop the observed profiles from PHC (at 1000 m), there is potential for much better agreement by systematically adjusting the representation of the local bathymetry and the width of the strait. In order to simulate the correct spreading of Mediterranean Waters from the Gulf of Cádiz into the North Atlantic, another approach could be to add additional physics like the effect of tides (Izquierdo et al., 2016), which are usually not included in current climate models. Without tides, ocean models often simulate erroneous south-westward spreading, leading to stronger biases when compared to climatology than in simulations with active tides (Izquierdo and Mikolajewicz, 2018).

#### 3.4.2 Surface conditions

Since there are no heat sources or sinks in the interior ocean, the observed deep bias cannot develop in-situ. Furthermore, since there is no sizable cold (fresh) bias above 1000 m, it cannot be entirely explained by a vertical redistribution of heat (salt). Instead, the surface has to be a major origin of the simulated deep ocean warming, and improvements in the deep ocean hydrography with higher resolution should be caused by improved surface fields.

Focusing on the SST bias in the last 30 years of the REF, MR0, and HR preindustrial simulations (years 71–100) in detail (Fig. 9), systematic differences between the simulations are evident (for the discussion of LR and MR, see Appendix A). The surface is consistently colder than PHC in all simulations, which is expected, since pre-industrial (PI) runs are compared with a

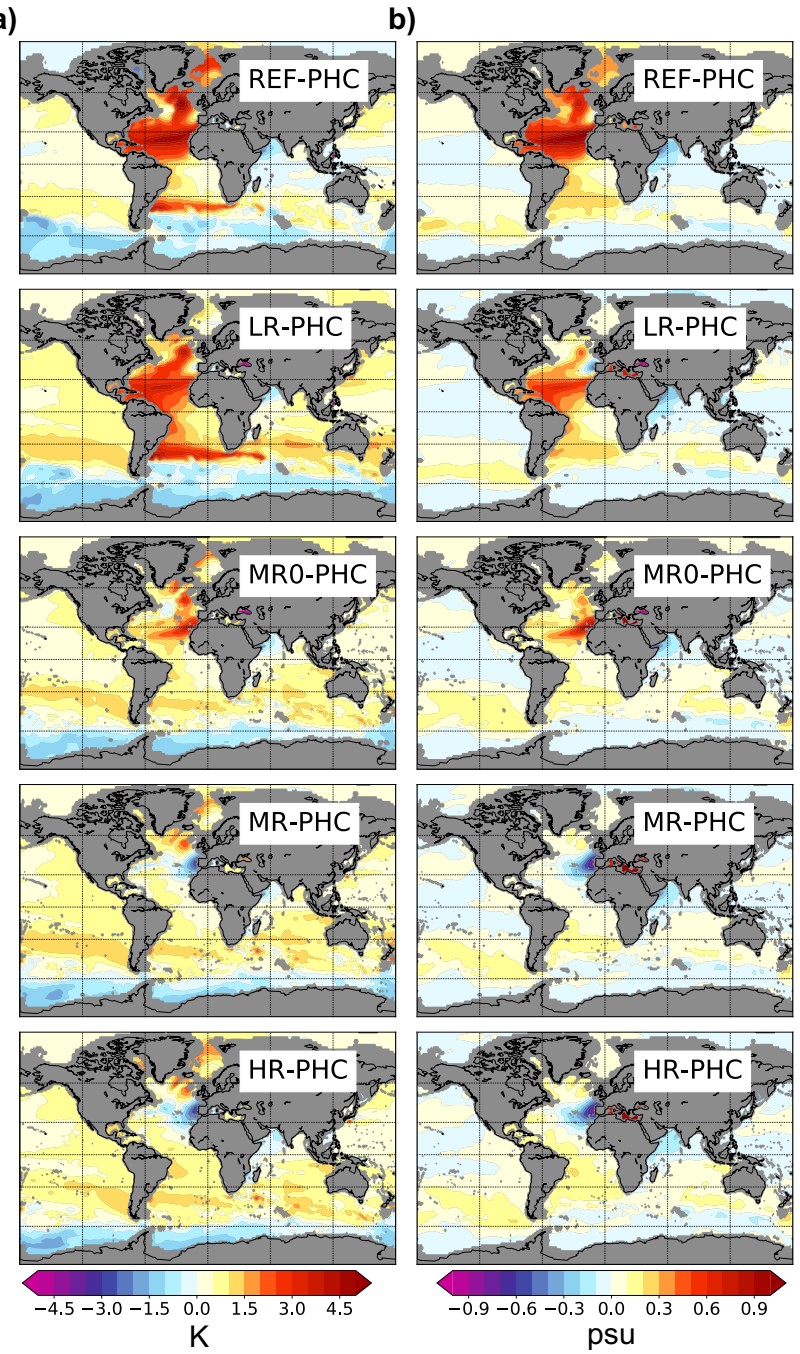

**Figure 7. a)** Potential temperature [K] and **b)** salinity [psu] biases with respect to the PHC climatology (PHC 3.0, updated from: Steele et al., 2001) at 1000 m depth, plotted on the observational grid. A systematic decrease of the temperature and salinity biases in the North Atlantic with increasing resolution (top to bottom) is evident.

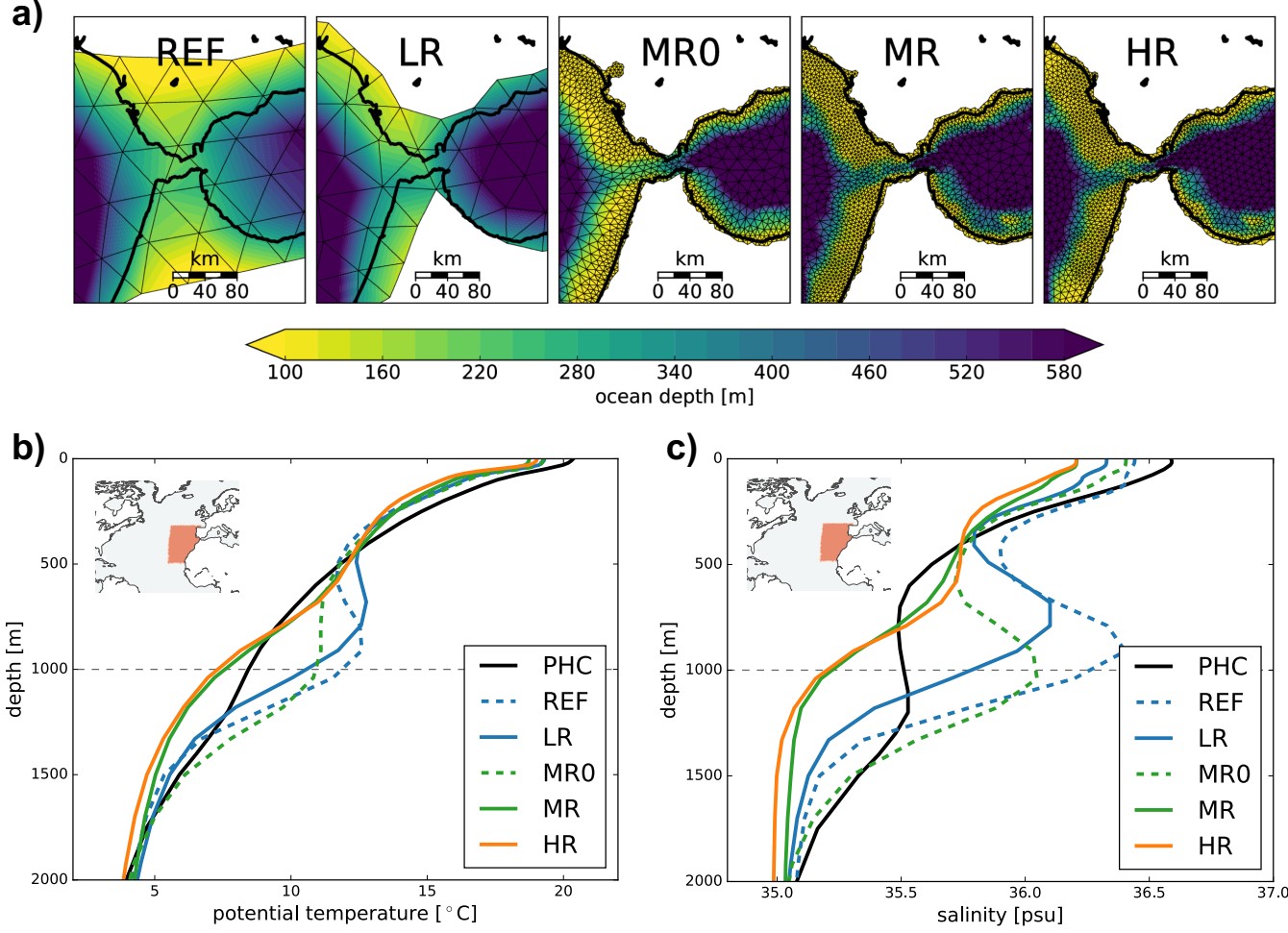

**Figure 8. a)** Spatial discretization of the Strait of Gibraltar and the Gulf of Cádiz in the five different model grids. The thick black line shows the true coastline as implemented in the Basemap plotting toolbox, using data from GSHHS (http://www.soest.hawaii.edu/pwessel/gshhs/index.html, last access: 17 May 2019). Triangular elements are shown with thin black lines, colors depict the local ocean depth in meters. **b) and c)** Vertical profiles of regionally-averaged potential temperature and salinity in the vicinity of the Strait of Gibraltar ($5°$W–$30°$W and $20°$N–$40°$N; red box in the insets). The horizontal dashed line highlights the depth of $1000$ m.

climatology representing present-day conditions. However, in the whole Labrador Sea, REF, MR0, and HR are on the warmer side for years 71-100. When overlaying their SST bias with simulated surface isopycnals (gray and black contours in Fig. 9b–d), which represent the mapping to the deep ocean in 600–1000 m depth (see details in the sections below), it is evident that warm SSTs over these critical regions are systematically reduced when going to the higher resolutions (Fig. 9b–d). Consistent with uncoupled ocean-only results for LR and HR (Sein et al., 2016, their Fig. 7)), which show a much better simulation of the position and separation of the Gulf Stream further south at higher resolutions, the coupled simulations analyzed here also show a successively reduced meridional warm/cold bias pattern along the East Coast of North America.

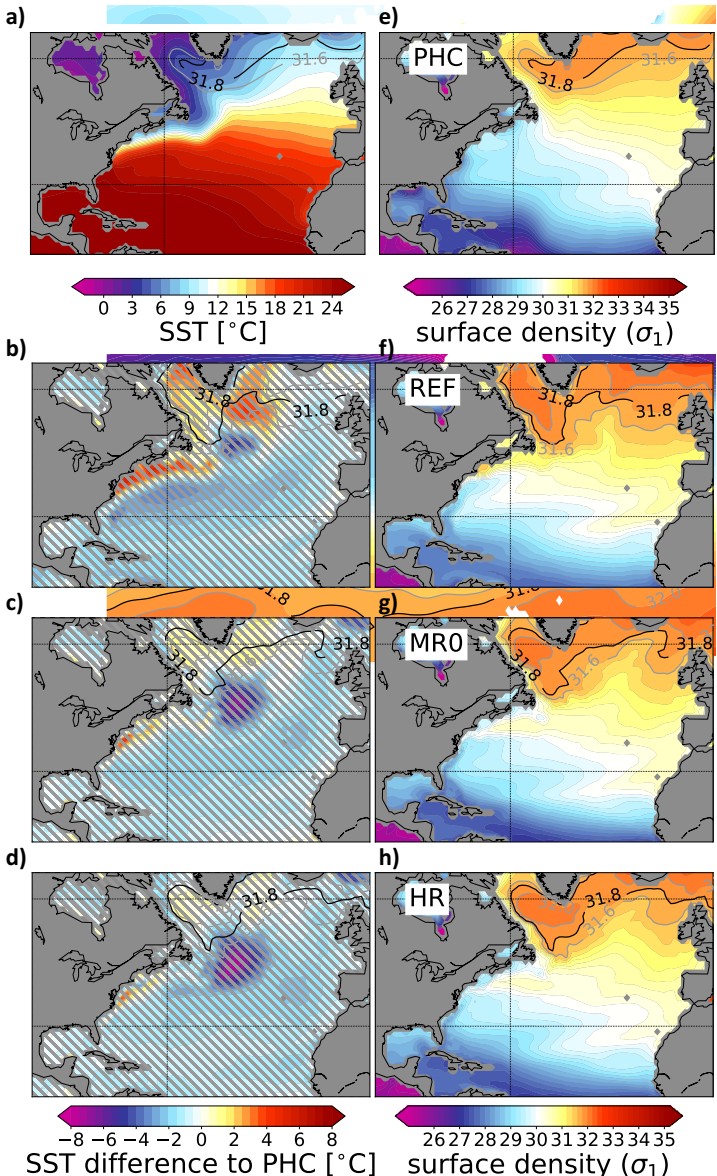

**Figure 9. a)** The North Atlantic sea surface temperature and **e)** $\sigma_1$-density structure at the ocean surface, as determined from the PHC climatology (PHC 3.0, updated from: Steele et al., 2001). Black and gray contours indicate outcropping areas for typical isopycnal surfaces found in the deep ocean around 1000 m (e.g. $\sigma_1 = 31.8$). **f)–h)** Same for the simulated density structure in REF, MR0, and HR (years 71–100). **b)–d)** Sea surface temperature (SST) biases in the 3 simulations (years 71–100) with respect to PHC in Fig. 9a. Three simulated $\sigma_1$-density contours that represent the 'mapping' to deeper ocean layers are overlaid with black and gray contours (identical to the contour levels in Fig. 9f–h). To highlight the SST improvements in the areas encircled by these contours, hatching grays out regions that are not in contact with the deep ocean around 600–1000m (based on the 30-yr annual means).

Despite these clear improvements over the deep convection sites and over the Gulf Stream region, the cold temperature spot in the North-West corner is a persistent bias and is even better visible in the medium- and high-resolution coupled simulations, since the surrounding warm biases are much reduced. Note that also uncoupled ocean-only models still struggle to properly simulate the North-West corner of the North Atlantic (Sein et al., 2017), and presumably much higher resolution along with a more detailed representation of the bathymetry is needed for the Gulf Stream to reach this area. Although the Gulf Stream and its extension could impact the location of the outcropping regions, the strong cold temperature spot (hatched in Fig. 9b–d) is, however, not in direct contact with the deep ocean around 600–1000 m depth via outcropping isopycnals (as diagnosed from 30-yr annual means). Despite possible seasonal excursions, we therefore do not expect a major impact on the analysis of the present study, which is focused on the deep ocean.

We conclude that the deeper ocean is connected to less warm surface conditions (non-hatched regions in Fig. 9b–d) in the higher resolution model versions, and in the next section we will study how this translates to the improvements seen in the deep ocean.

## 3.5 Along-isopycnal bias propagation in the Atlantic

By focusing on surfaces of constant potential density (isopycnals), it is possible to trace the development of the biases from the surface to the deep ocean around 1000 m depth, where our lower-resolution simulations and the CMIP5 models show the strong anomalous warming (Fig. 1). We compute running 10-yr means for the temperature bias along the $\sigma_1 = 31.8$ isopycnal ($\sigma_1$ denotes potential density, referenced to 1000 m depth). We chose this specific isopycnal, because it coincides with a depth of 800–1000 m in the North Atlantic area (Fig. 10). It also lies in the middle of the envelope formed by the 31.6 and 32.0 contours that were already shown in Fig. 9 (gray contours).

To isolate the influence of the chosen ocean grid using the same atmospheric T127 configuration, we will focus on the LR and HR configurations here as examples. When looking at the bias development in LR (see animation S1 in the video supplement (Rackow et al., 2018b)), there are two major surface source regions for the deep bias in the Atlantic—the Strait of Gibraltar and the north-eastern North Atlantic. The first source of the warm bias in 1000m is likely to be of geometric nature, since the very narrow Strait of Gibraltar cannot be properly discretized at coarse resolutions. However, simply increasing the resolution in the Strait of Gibraltar does not automatically remove the bias; instead, climatological T/S profiles in the vicinity of Gibraltar lie between the according REF/LR/MR0 and MR/HR profiles (Fig. 8b,c). As mentioned before, a systematic geometric tuning of the ocean bathymetry in this area was not attempted (Fig. 8a), and there is thus potential for closer agreement with climatological potential temperature and salinity profiles in this region by adjusting the spatial resolution within (and in the vicinity) of the strait.

The source in the north-eastern North Atlantic is related to enhanced downwelling and an erroneously deep mixed-layer ($\geq 500$ m; green contours in the supplemental animation) in this area. This is a feature that has already been identified in uncoupled FESOM simulations using the LR grid as part of the CORE-II intercomparison project (Danabasoglu et al., 2014, their Fig. 13). Since the Gulf Stream in the LR (and REF) simulations is too zonal and reaches the northeastern North Atlantic, part of the flow has to downwell here, which we suspect could explain part of this deficiency by entraining waters and deepening

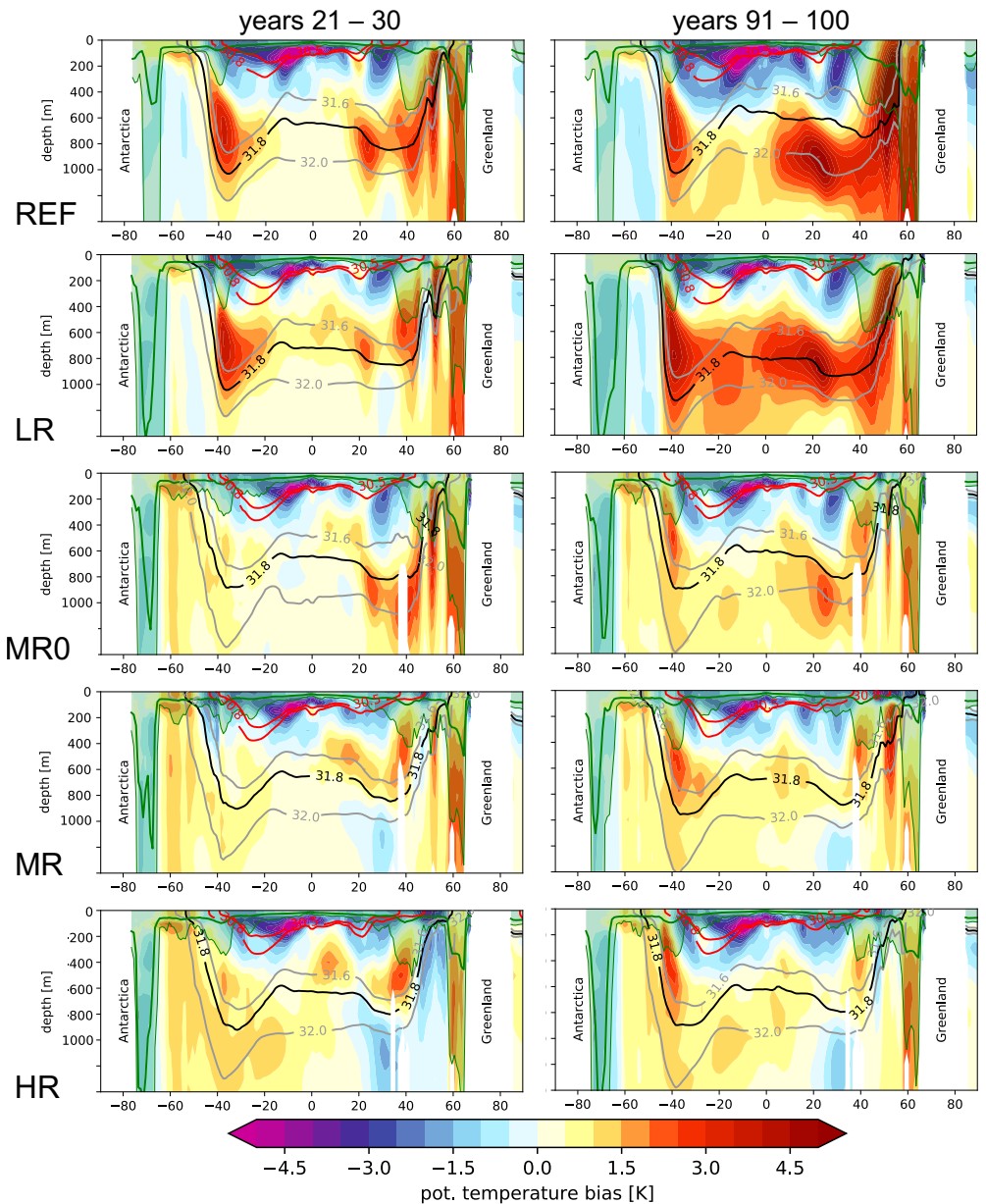

**Figure 10.** Meridional section at 30.5 °W through the Atlantic Ocean for the potential temperature bias in the five simulations. The difference compared to the PHC climatology is shown with colours for years 21–30 **(left column)** and years 91–100 **(right column)**, illustrating the North Atlantic bias development along isopycnal layers (see animations S3 and S4 for LR and HR with a 10yr running window in the video supplement (Rackow et al., 2018b)). The contours show $\sigma_1$ density contours that are representative for the deep ocean between 600 and 1000 m (gray and black; $\sigma_1 =$31.6, 31.8, 32.0) and for the surface ocean until a maximum depth of about 300 m (red; $\sigma_1 =$30.5, 30.8). The average (maximum) mixed layer depth in the 10-yr windows is overlaid with a green line (green shading).

the mixed layer. Other factors influencing the mixed layer depth could be biased buoyancy fluxes or the restratification process via eddy activity. By comparing the years 21–30 to the years 91–100 of LR (second row in Fig. 10) the advective nature of the bias signal propagation from the surface in high latitudes to the deep ocean at lower latitudes is evident, which coincides with the mean currents of the subtropical gyre that go into the same direction.

The mixed layer (green line and shading in Fig. 10) is deep enough so that surface biases can reach the 31.8 and neighboring isopycnals, from where the signal is further advected towards the south. Eventually the signal is advected towards the equator, from where it propagates to the East as a Kelvin wave (video supplement S1).

    In contrast, all above mentioned issues are almost absent in the HR configuration (see last row in Fig. 10 or the animation S2), which is a major improvement compared to the previous AWI-CM-LR configuration. This strongly suggests that also in
the CMIP5 models the lack of spatial resolution is favouring biases in the deep ocean. Higher spatial resolution is needed to properly resolve the very narrow geometry of the Strait of Gibraltar and it is one way to better simulate the position of the Gulf Stream, although other factors also play an important role. The latter improvement reduces warm SST biases over North Atlantic areas that are in contact with the deeper ocean (Fig. 9b–d), which in turn reduces the warming in the deep ocean. While a strong resolution-dependence was also shown by Marzocchi et al. (2015), there are additional ways for getting a more
realistic Gulf Stream separation. These include details of the numerical scheme that can affect current-topography interactions (Penduff et al., 2007) or the representation of non-local dynamics that impact the formation of a northern recirculation gyre along the North American coast, such as the Deep Western Boundary Current downstream of Cape Hatteras (Zhang and Vallis, 2007) and the cold Labrador Current northward of the Gulf Stream front (Sein et al., 2017).

### 3.6   Displacement and tilt of simulated isopycnals

There is a third source of biases, which is responsible for the deep ocean warming in the Southern Ocean. It is related to the fact that the eddy parameterization (GM) has difficulties in representing the slope of the isopycnals, which is determined by the counteracting effects of Ekman pumping and eddy transport (Farneti et al., 2015). As an example, meridional sections along 10.5°E reveal that the strong deep ocean warming in LR seen in Fig. 7a to the West of Cape Agulhas is linked to too steep simulated isopycnals between 40°S and 45°S (black and grey contours in Fig. 11b, left) compared to the much flatter observed
tilt of the isopycnals (as in PHC; magenta contours). Already at medium resolution (MR), the simulated isopycnal slope is about halved compared to LR and much closer to the observed slope (Fig. 11, right) with strongly reduced temperature biases, suggesting that the explicitly resolved eddies outperform the eddy parameterization as applied in the prototype simulations with AWI-CM (using a default $K_{GM}$).

    Isopycnals in the upper ocean above 200–300 m in MR ($\sigma_1 = 30.5, 30.8$) are also much closer to the observed state from PHC
than in LR (compare red contours to magenta contours in Fig. 11b), associated with an interior bias dipole of warmer/colder temperatures in LR (left panel) and a more homogeneous (cold) bias pattern in MR (right panel). Interestingly, the surface representation (SST bias) of this warm/cold interior bias to the west of Cape Agulhas and a similar dipole-like bias in the Brazil-Malvinas Confluence region are cleanly separated into their warm and cold parts by the $\sigma_1 = 30.5$ isopycnal surface contour (red contour in Fig. 11a, left) in LR. This suggests that these biases could be caused by shifted water masses as indicated

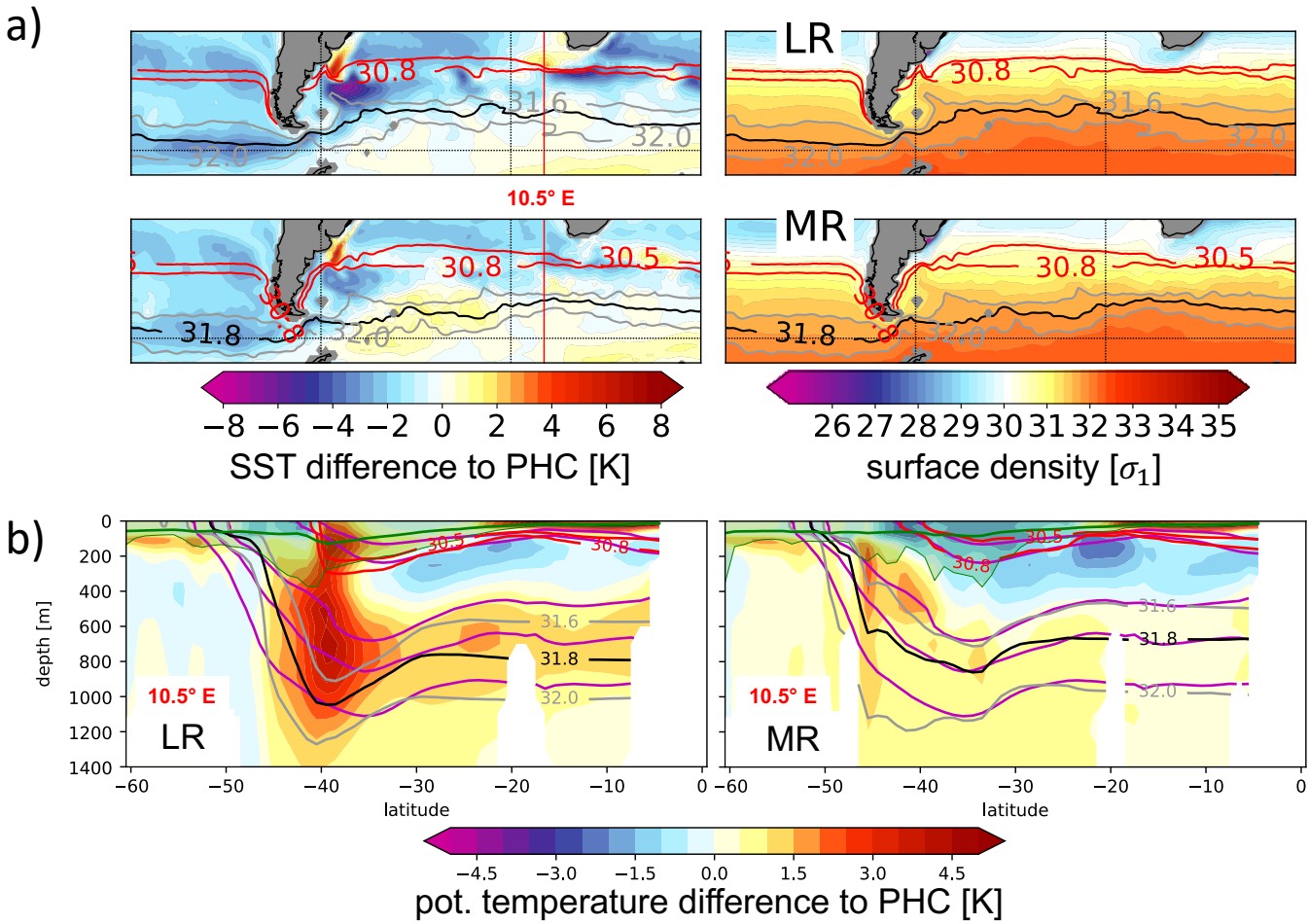

**Figure 11. a, (right)** The Southern Ocean $\sigma_1$-density structure at the ocean surface in LR and MR (years 71–100). Black and gray contours indicate outcropping areas for typical isopycnal surfaces found in the deep ocean around 1000 m; red contours represent shallower isopycnal surfaces with a maximum depth of about 200 m. **a, (left)** Sea surface temperature (SST) biases in the 2 simulations (years 71–100) with respect to PHC. Simulated $\sigma_1$-density contours are overlaid (identical to the contour levels in the **right panels**). A meridional section at 10.5 °E is highlighted with a vertical red line. **b)** Meridional section at 10.5 °E, to the west of Cape Agulhas, showing the potential temperature bias with respect to PHC in **(left)** LR and **(right)** MR (years 71–100). Contours show simulated $\sigma_1$-density contours that are representative for the deep ocean between $\approx$600 and 1000 m (gray and black; 31.6, 31.8, 32.0) and for the surface ocean until a maximum depth of about 200 m (red; 30.5, 30.8). In contrast to LR, the tilt of the isopycnals in MR is a close fit to the 'target' $\sigma_1$-contours from PHC (given in magenta). The average (maximum) mixed layer depth in the 30-yr window is overlaid with a green line (green shading).

by the erroneous northward shift of the $\sigma_1 = 30.5$ contour, leading to a warm bias on its northern side and to a cold bias on its southern side. Flattening the slope would result in a southward shift with potentially reduced biases. Indeed, the surface biases

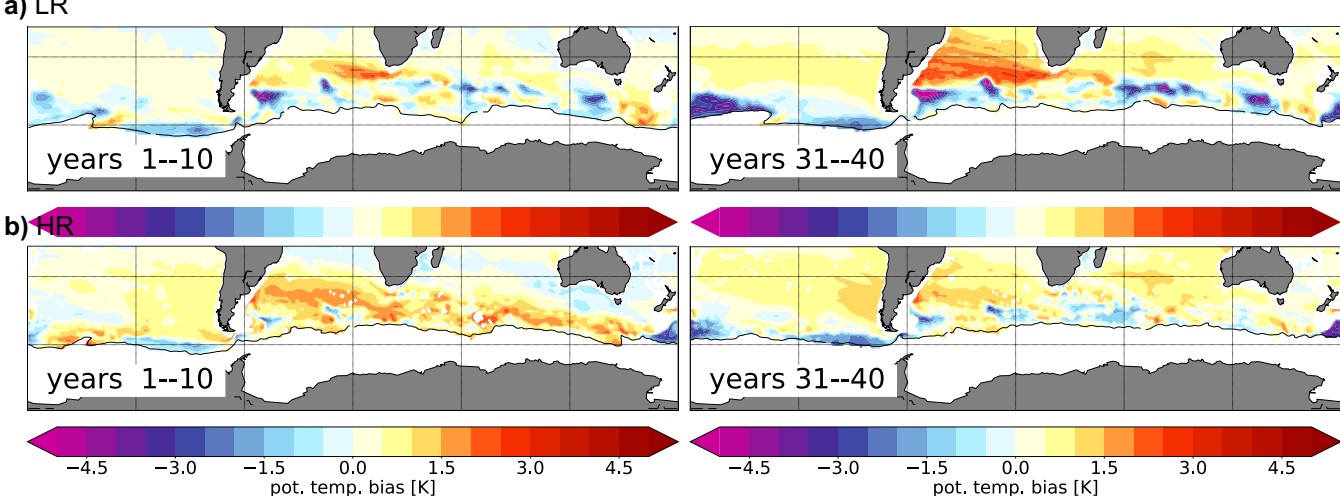

**Figure 12.** Southern Ocean potential temperature biases [K] with respect to PHC, on the constant isopycnal $\sigma_1$=31.8, in **a)** LR and **b)** HR for years 1–10 and 31–40. Black contours show the outcropping location of the $\sigma_1$=31.8 isopycnal. Areas to the south of the outcropping location are white (indicating no data). For animations of the bias development with a 10-yr running window, see supplementary animations S1 and S2 (Rackow et al., 2018b).

are strongly diminished in MR (Fig. 11a) with better resolved eddies and the associated flatter isopycnals, which are a close fit to the target contours from PHC (Fig. 11b).

## 4   Perspective and implications for model initialization

The five simulations in this study are initialized from rest with zero velocities, prescribing long-term mean temperature and
salinity fields for boreal winter from PHC. This leads to a fast initial adjustment of geostrophic currents, usually based on a rather smooth climatology as done in this study, while in reality, e.g., zonal fronts will move up and down throughout the year. After this first phase of fast adjustment, which takes months to one year and is also influenced by the topography as represented on the model grids, significant biases are already apparent after the first years (not shown). As an example, the warm/cold bias pattern along the eastern coast of North America (Fig. 9b), which is related to the too northerly course of the Gulf Stream in
AWI-CM-LR and REF, fully develops within a couple of years. We are confident that a focus on (and good understanding of) the initial bias development could lead to significantly improved models, as the later stages are likely dominated by slow developments in the deep ocean, following these fast initial 'damages'. At higher resolutions like MR and HR, when initialized from zero velocities, it could also become important to temporarily hold the 3D temperature and salinity fields close to a (seasonally varying) climatology as the circulation and eddy fields are still developing.
Interestingly, in HR, an initial movement of the 31.8 isopycnal surface contour in the Southern Ocean towards the equator apparently leads to larger initial biases than in LR (Fig. 12, left), and then it returns back to the south after 20 or 30 years. In

years 31–40, the biases seem to recover and are again smaller than in LR (Fig. 12, right). We hypothesize that this is (i) due to the westerly winds that quickly steepen the isopycnals, thus increasing baroclinicity; and (ii) due to the slowly developing eddy field that later flattens the isopycnals, which again shifts the outcropping region back towards the south. The time scale for the development of the Southern Ocean eddy field is several tens of years (Allison et al., 2010), which fits the behaviour described above. In contrast, the eddy parameterization in LR is active from the start, which keeps the isopycnals initially closer to the observed state, only to be outperformed by the HR simulation with explicitly resolved eddies in the later stages of the simulation.

## 5 Conclusions

It has been found that CMIP5 models tend to show a strong anomalous warming and salinization in the deep North Atlantic Ocean. Although being substantial in magnitude, to our surprise the deep ocean biases in CMIP5 models did not receive a lot of attention yet. While one could argue that this bias is 'well-hidden' from the atmosphere and therefore not as critical for climate simulations as surface biases, it can impact the outcropping and position of isopycnals. This could lead to a wrong mapping of the deep ocean to the surface and as a consequence to erroneous projections of the heat uptake of the deep ocean. Here we exploit the fact that the AWI-CM at low CMIP5-type resolutions reproduces the behaviour seen in CMIP5 models. We show how the deep ocean bias develops from the surface and how it propagates along relevant isopycnal layers into the deep ocean. Along-isopycnal analyses are common oceanographic diagnostics to trace sources and pathways of temperature and salinity anomalies (e.g. Alban et al., 2001; Nonaka and Sasaki, 2007); and they could be further applied in climate models to determine pathways of anthropogenic heat uptake by the ocean. While the improvements are not strictly monotonic, we found that the deep bias seen in AWI-CM-LR and REF is generally reduced when moving to higher resolutions (10 km and higher) in eddy-active regions, a capability supported by FESOM1.4's use of multi-resolution ocean grids. Although there is certainly scope for improved eddy parameterizations, our results thus highlight the benefit of using high-resolution ocean components in climate modelling.

It should be mentioned that the flexibility of unstructured multi-resolution ocean grids comes with its own challenges: How best to distribute a given number of computational grid points over the globe in climate simulations? While in the past, more idealized approaches to the distribution of the spatial resolution have been performed at AWI (e.g. the refinement towards 0.25° along the equator in REF, or resolution increases over the whole Arctic in LR), the medium- and high-resolution meshes follow a more objective global strategy by focusing resolution in regions of strong observed eddy variability. As a consequence, for example the nominally coarsest mesh, REF, features the highest resolution in the tropical Pacific Ocean among all meshes. Despite the fact that the resolution change in the five meshes is thus not strictly systematic over the global ocean, there is a systematic increase of spatial resolution in the North Atlantic. Since we only consider 100-yr simulations in this study, we do not expect resolution changes in the other basins to impact the simulation of the North Atlantic and the conclusions of our study.

Potentially, the chosen vertical mixing scheme could also impact biases in the deep ocean. However, we could not identify a clear dependence of deep ocean biases on the vertical mixing schemes used in CMIP5 models: the three models with the strongest absolute error at a depth of 1000 m (GISS-E2-R, MPI-ESM-LR, GFDL-CM3; see mean absolute potential temperature error in 1000 m in Table 1) use either KPP or PP mixing (Huang et al., 2014, their Table 1). This suggests that spatial resolution provides an alternative way to reduce long-standing deep ocean biases.

We identified two major sources for the deep ocean biases in the Atlantic ocean. The first source is the Strait of Gibraltar, which is likely to be a geometric issue related to the spatial discretization of this narrow strait (15 km) at relatively coarse resolution that is typical for CMIP5 models (about 100 km), and that often leads to increased Mediterranean outflow (e.g., Sterl et al., 2012). Much more systematic efforts are required to tune the horizontal and at the same time the vertical discretization of the Strait of Gibraltar. The warm and saline biases originating from this area largely disappear with higher resolution in AWI-CM-MR/HR, probably due to lower spurious numerical mixing and an improved representation of the bathymetry to the West of the Strait of Gibraltar (Fig. 8), which should add to the realism of the simulated plume. At the highest resolutions considered here, the bias in the proximity of the Strait of Gibraltar changes sign towards a too cold and fresh anomaly. Ongoing tests suggest a similar sensitivity to the chosen vertical viscosity/diffusion, as it can also affect the exchange by changing the friction between Atlantic and Mediterranean waters (not shown). We suspect that besides local resolution increases using multi-resolution grids, the incorporation of (the effect of) tides in climate models and the addition of an overflow parameterization might be necessary steps to further improve the model performance.

The second source in the low-resolution configurations is the north-eastern North Atlantic, where erroneous downwelling associated with typically anomalously deep MLD (Danabasoglu et al., 2014, their Fig. 13) communicates biased surface conditions into deeper layers. The signal then further propagates along isopycnal layers with the sub-polar gyre circulation into the deep Atlantic around 1000 m. This source of the deep ocean biases is largely diminished in the higher resolution configurations, which better simulate the separation of the Gulf Stream and the North Atlantic Current; and, in fact, we could ascribe the improvement in the deep ocean to smaller SST biases over ocean regions that are in contact with the deeper layers around 1000 m.

In the Southern Ocean, there is a third source of deep ocean warming that is related to a displacement of isopycnals, which are locally too steep on the coarse meshes with active default eddy parameterization. Thus, outcropping often happens too far to the north compared to observations, so that denser water masses will be in contact with atmospheric conditions (fluxes) that are usually in contact with lighter waters, which can impact water mass transformation. Compared to parameterized eddies (with the default GM coefficient), explicitly resolved eddies in the prototype simulations tend to flatten the isopycnals stronger, which reduces sub-surface biases as well as their surface representations locally, e.g. to the West of Cape Agulhas and in the Brazil-Malvinas Confluence region. Since we were using a default GM coefficient for all simulations, it can be argued that a regional tuning of GM with a horizontally varying coefficient (Visbeck et al., 1997; Danabasoglu et al., 2012) could lead to a better simulation of the Southern Ocean in low-resolution AWI-CM configurations. Moreover, high-resolution simulations and their effective $K_{GM}$ could also serve as a template for the regional tuning of low-resolution simulations.

The remaining biases between $\pm 20$–$40°$N/S, seen in meridional sections along $30.5\,°$W through the Atlantic, show a consistent warm/cold pattern in the vertical direction. Griffies et al. (2015) also study surface and interior temperature bias maps and show that "where the upper portion of the gyres is cool, the deeper portion is warm". They conclude that mean vertical heat transport from the upper ocean into the interior ocean by the time-mean currents is too strong in their $1°$ (and to some extent in their $0.25°$) configurations, or rather it is not sufficiently compensated by the upward transport from mesoscale eddies. Apparently, typical current eddy parameterizations are not sufficient to offset the downward heat transport from the mean circulation. This implies a possible limitation of our focus of high spatial resolution only in areas of strong eddy activity in AWI-CM-MR and -HR (mainly over the western boundary currents and in the Southern Ocean) since resolution could be important even in the gyre centers to get a realistic magnitude of vertical eddy transports.

The Hovmoeller diagrams for the potential temperature and salinity in the North Atlantic Ocean reveal strongly reduced drifts in the interior ocean at medium and high resolutions, which fits previous findings (von Storch et al., 2016; Hewitt et al., 2017). However, one cannot rule out the possibility that the higher resolution configurations could be drifting only slower towards an equally large equilibrium error, and it remains to be seen whether the strong improvements seen over the 100yr-timescale will last on multi-centennial timescales. Even so, a slower drift at higher ocean resolution is certainly very beneficial for efforts related to ocean reanalysis, and seasonal, interannual, and decadal prediction.

Overall, we have shown major improvements when using medium-resolution (MR) and high-resolution (HR) meshes on representing the hydrography in the deep ocean around $1000\,$m. These improvements at depth do not come at the expense of degradations in other climatically relevant fields, as shown by a performance index analysis (Appendix B), but rather improve both the ocean and atmospheric simulation. These grids are partly eddy-resolving and partly at most eddy-permitting, so that eddy parameterizations still need to be applied locally. This calls for dedicated in-depth analyses of eddy heat fluxes (and budgets) and their representation on multi-resolution unstructured grids in future studies. Owing to the competitive speed of 6 simulated years per day, the MR mesh can be used for our CMIP6 standard configuration AWI-CM-MR (with T127 atmosphere), and the HR mesh is used in the HighResMIP project. Next steps will be the development of frontier climate simulations (e.g. AWI-CM-XR) with meshes of 6 million (or more) surface grid points and higher-resolution atmospheres (T255 or higher). With FESOM1.4's finite-volume successor FESOM2 (Danilov et al., 2017), which is $\sim 3$ times faster and more resource-efficient, running this class of flagship meshes will become possible even for coupled simulations. The corresponding coupled model with its tentative name AWI-CM2 is close to its test phase, and we expect a major step change in the quality of the simulated climate at these resolutions.

This paper does not document AWI's final CMIP6 pre-industrial control simulations (Semmler et al., 2018) that will undergo additional changes to the model configuration and further tuning. Tuning could potentially affect the deep ocean simulation, although the global top-of-the-atmosphere (TOA) balance in particular appears not to be directly related to the magnitude of North Atlantic deep ocean biases (not shown). Additionally, the final simulations will use updated ozone forcing that had not yet been available at the time of writing. However, we deem it very important to report on significant improvements during the model development cycle that could also be of interest for other groups developing high-resolution models, in order to document identified sensitivities of model biases to the various possible sources in global coupled climate models.

## 6 Code availability

The source code and used configuration (namelists) for the coupled FESOM model that is part of AWI-CM1.0 is archived at http://doi.org/10.5281/zenodo.1342014 (Rackow et al., 2018a). The ECHAM6 source code is maintained by the Max Planck Institute for Meteorology and freely available to the public at large (http://www.mpimet.mpg.de/en/science/models/mpi-esm/echam/). External access to the ECHAM6 model is provided through their licensing procedure (http://www.mpimet.mpg.de/en/science/models/license/). If you are interested in the full coupled model including the ECHAM6 sources, you need to register on the MPI-ESM user page (https://www.mpimet.mpg.de/en/science/models/mpi-esm/users-forum/) and then download the complete coupled AWI-CM model (rev140 was used in this study) from the SVN repository at https://swrepo1.awi.de/svn/awi-cm/trunk@140. After registering, the code can be accessed using the open-source subversion software (http://subversion.apache.org/). Updated code for AWI-CM will be available through the same link. Mesh partitioning in FESOM is based on the METIS Version 4.0 package developed at the Department of Computer Science & Engineering at the University of Minnesota (http://glaros.dtc.umn.edu/gkhome/views/metis). METIS and the pARMS solver (Li et al., 2003) are separate libraries which are freely available subject to their licenses. The OASIS3-MCT coupler is available for download at https://portal.enes.org/oasis.

## 7 Data availability

The video supplements S1 to S4 are archived at Zenodo, http://doi.org/10.5281/zenodo.1323334 (Rackow et al., 2018b). The data of the five simulations (years 71–100) can be publicly accessed at the DKRZ cloud at https://swiftbrowser.dkrz.de/public/dkrz_035d8f6ff058403bb42f8302e6badfbc/Rackow_DeepBias_GMD2018/. The Polar Science Center Hydrographic Climatology (PHC3.0; Steele et al., 2001) is used for comparison and is freely available online (http://psc.apl.washington.edu/nonwp_projects/PHC/Data3.html).

## Appendix A: Surface conditions in LR and MR

The applied model version of AWI-CM (rev140) has too high simulated variability in the Labrador Sea, causing occasional "on" and "off" episodes of deep convection in the Labrador, which can mask changes at the surface on a decadal time-scale (Sidorenko et al., 2015; Rackow et al., 2016). In the whole Labrador Sea, LR and MR show only cold SST biases (not shown) for years 71-100, while the other three configurations (REF, MR0, HR) are on the warmer side (Fig. 9b–d). As mentioned above, the LR and MR behavior can be explained by the occurrence of strongly reduced deep convection in those years (green and blue solid lines in Fig. 13) associated with too high sea-ice coverage, leading to the strong cold SST biases. To draw definite conclusions at the surface for the LR and MR configuration is thus more difficult than for the deep ocean analysis. We therefore focused the surface analysis in section 3.4.2 on the other low-, medium-, and high-resolution simulations (REF, MR0, and HR). A separate branch of development at AWI is dealing with this issue of too high variability in the Labrador Sea, and in preliminary tests with a newer AWI-CM version that uses a different mixing scheme in the ocean (KPP; Large et al., 1994)

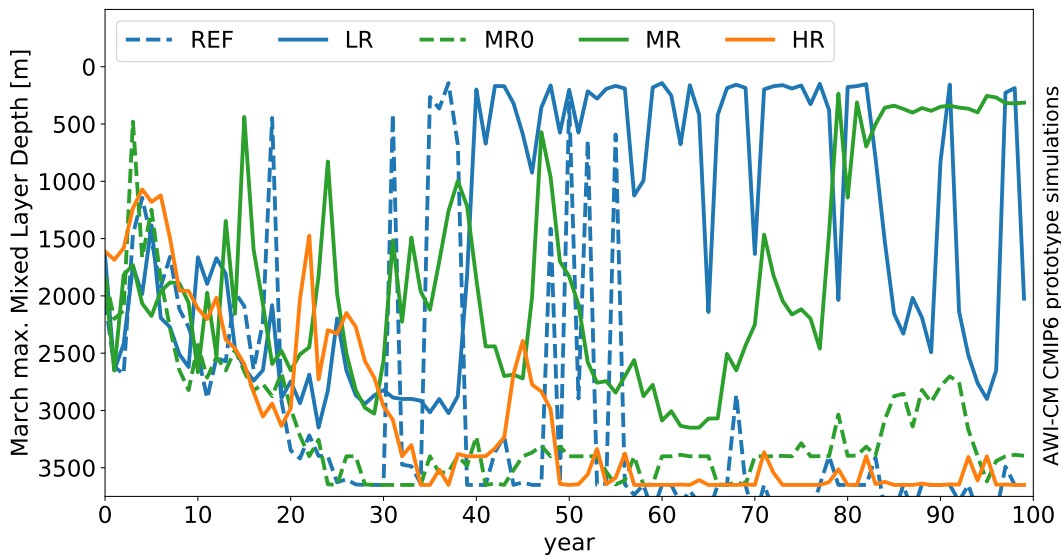

**Figure 13.** Maximum mixed layer depth [m] for March in the Labrador Sea for the five 100-yr simulations with AWI-CM. The simulated mixed layer starts to diverge after about 20-30 years into the coupled simulations. At the end of the simulation (years 71—100), LR and MR have the lowest mixed-layer while REF, MR0, and HR simulate overly deep mixed-layers in the Labrador Sea.

and newer versions of ECHAM6 (ECHAM 6.3.02p4/6.3.04p1) this issue is gone, and we will report on these simulations in the future.

## Appendix B: Computation of oceanic performance indices

Extending on the idea to compute performance indices (PI) that grade climate model simulations of various atmospheric parameters (Reichler and Kim, 2008), performance indices for the ocean are computed in this study as follows: First, FESOM potential temperature and salinity data are interpolated horizontally and vertically to the grid of the PHC climatology. This is done for both climatological winter (DJF) and summer (JJA) means of the last 30 years of the AWI-CM simulations. Afterwards, the absolute winter and summer temperature and salinity errors with respect to the PHC climatology are calculated for each grid point and averaged globally, or over individual ocean basins. The same is done with an ensemble of 21 CMIP5 models for which the three-dimensional temperature and salinity fields were available at the time of download. FESOM absolute errors for winter and summer temperature and salinity are normalized with the mean absolute errors of the CMIP5 ensemble (for each individual ocean basin and globally). In Table 4, we give the average over the two parameters and two seasons globally and for two key ocean areas (North Atlantic and Southern Ocean). We set the southern limit of the North Atlantic as 0°N while the northern limit is composed of the 65°N latitude line west of Iceland, a straight line from Iceland to Spitsbergen, and a straight line from Spitsbergen to the northern tip of Norway (as shown in Fig. 5f). The Southern Ocean is defined here as the ocean area

**Table 4.** Oceanic performance indices (PI) for the global ocean, two important areas (North Atlantic and Southern Ocean), and PI for key atmospheric parameters

| AWI-CM configuration | Oceanic PI | | | Atmospheric PI* |
|---|---|---|---|---|
| | Global Ocean | North Atlantic | Southern Ocean | |
| REF (T63) | 0.87 | 0.98 | 0.68 | 1.03 |
| LR (T127) | 0.72 | 0.80 | 0.74 | 0.87 |
| MR (T127) | 0.64 | 0.62 | 0.62 | 0.81 |
| MR0 (T127) | 0.61 | 0.57 | 0.62 | 0.79 |
| HR (T127) | 0.66 | 0.63 | 0.62 | 0.80 |

*PIs below (above) 1 indicate that a model performs better (worse) than the average of the considered CMIP5 models
(Sidorenko et al., 2015)

south of 40°S. The atmospheric PI are computed as detailed in Sidorenko et al. (2015) and Appendix 1 and 2 in Rackow et al. (2016).

*Author contributions.* TR conceived the study together with SD and TJ. TR performed the analysis and wrote the manuscript with contributions from all co-authors. DSe, TS, and TR ran the pre-industrial simulations. TS performed the performance index analysis and created
Figure 1. NK and DSi supported the post-processing of the data. All authors added to the scientific discussion.

*Acknowledgements.* D. Sein's work was supported by the PRIMAVERA project, which has received funding from the European Union's Horizon 2020 research and innovation programme under grant agreement No 641727, and by the state assignment of FASO Russia (theme 0149-2019-0015). N. Koldunov benefited from funding by the German Research Foundation through the Collaborative Research Centre TRR 181 "Energy Transfer in Atmosphere and Ocean". D. Sidorenko and Q. Wang were supported through the regional climate initiative
REKLIM. Some of the research underlying this report was also funded by the Federal Ministry for Education and Research (Germany) with the support code 01DJ15029 and 01DJ16016. The responsibility for the content of this publication lies with the authors. All plots except Figure 2 have been made with the 2D graphics environment Matplotlib (Hunter, 2007) using Jupyter (Kluyver et al., 2016). We acknowledge the World Climate Research Programme's Working Group on Coupled Modelling, which is responsible for CMIP, and we thank the climate modeling groups (listed in Table 1 of this paper) for producing and making available their model output. All simulations have been performed
at the German Climate Computing Center (DKRZ).

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
