# Peer review of "Sensitivity of deep ocean biases to horizontal resolution in prototype CMIP6 simulations with AWI-CM1.0"

_Geoscientific Model Development, 2018_

## Referee Comment (RC1) · Anonymous Referee #1 · 26 Sep 2018

Review of "Sensitivity of deep ocean biases to horizontal resolution in prototype CMIP6 simulations with AWI-CM1.0" by Rackow and Co-authors

The manuscript looks into the role of increased horizontal resolution in select regions of the ocean component of a coupled model in addressing, i.e., reducing, some of the deep ocean temperature and salinity biases in the Atlantic and Southern Ocean Basins. The authors argue that the ocean biases develop primarily from the surface, propagating along related isopycnals to the deep ocean. Higher horizontal resolution in the outcrop regions of these isopycnals appears to reduce such biases at depth. Although it is an interesting piece of work, I find the analysis rather superficial and qualitative, for

example, relying on animations, rather than quantitative analysis. I recommend major revisions along the following lines:

1. The Introduction actually introduces some physical mechanisms based on several previous studies concerning how deep temperature and salinity biases can emerge. In particular, the role of vertical mean and eddy heat transports is mentioned. Unfortunately, the manuscript does not get back to these points until the last section, and more importantly does not present a quantitative analysis exposing the role of various mechanisms. I strongly think that budget analyses should be included in the manuscript, in particular, exposing the changes in vertical eddy transports with increased horizontal resolution.

2. The authors identify three regions for the source of deep biases. The first is the Strait of Gibraltar. I do not necessarily agree with the authors view that incorporation of tides will improve the representation of Mediterranean Outflow. The outflow / overflow processes require resolutions of order 10s of meters in the horizontal and meters in the vertical. Two possible solutions are an overflow parameterization and changes in the bottom / lateral topography at the outflow of the Strait of Gibraltar to minimize spurious entrainment. The second source is identified as the erroneous downwelling associated with anomalously deep mixed layers in the northeastern North Atlantic. This statement is not justified. How do you know that the downwelling is erroneous and that the mixed layer depths are anomalously deep? The third source is presumably related to a displacement of isopycnals which are identified as too steep when eddies are parameterized. First, the analysis is not quantitative and I do not really follow the argument. Second, this is likely due to the issues with the details of the mesoscale eddy parameterization used. A description of the parameterization as implemented in the model should be included. Furthermore, since the REF case is much cheaper, a couple of cases with modified versions of the parameterization could be tested as alluded to in the text. Incidentally, I am not sure what is meant by mean absolute error. Is this the root-mean-square (rms) error?

3. The text refers to higher resolution configurations as (regionally) eddy-resolving in various places. Are they? As far as I can tell, they are still mostly eddy-permitting. A definition of what is meant by eddy-resolving and spatial maps of eddy-permitting and eddy-resolving regions for each configuration should be included. The text says "resolving the Rossby radius", but that is not a quantitative statement. What is the physical justification for cutting of the eddy parameterization below 25 km resolution, knowing that the resolutions are mostly on the eddy-permitting side? Also, as far as I can tell, the number of vertical levels is not given in the manuscript.

4. I am unsure if all the cases represent an apples-to-apples comparison. Specifically, these are fully coupled, pre-industrial simulations. Changes in one component will undoubtedly introduce the need to retune the top-of-the-atmosphere (TOA) radiation budget. Please provide a table with the TOA values for each configuration. My point is that if the reduced bias cases show large negative TOAs in comparison to the REF case, then when the coupled model is retuned, then it is possible that the deep ocean biases will reappear. Additionally, please include comparisons of the Atlantic meridional overturning circulation (AMOC), Labrador Sea Deep Water formation / mixed layer depth, and the northward heat and salt transports to show that the reductions in the deep biases are not occurring at the expense of degradations in several other climatically important fields.

5. In the last paragraph of section 3.4, it is stated that "higher spatial resolution is needed . . . . to better simulate the position of the Gulf Stream." I thought that there were studies in literature showing that the high resolution is not really the silver bullet. Perhaps an expanded discussion should be included here. Also, I do not really follow the argument made in the last paragraph of section 3.4.1.

---

## Referee Comment (RC2) · Anonymous Referee #2 · 28 Sep 2018

**"Sensitivity of deep ocean biases to horizontal resolution in prototype CMIP6 simulations with AWI-CM1.0" by Thomas Rackow et al.**

**#Referee 2**

Rackow et al. are describing a hierarchy of climate model using the AWI-CM. They present the capability of the ocean model on unstructured mesh for climate application. The focus is on the benefit of using local refinement in eddy active region to decrease the deep temperature bias. In addition to that, they discuss why the high resolution decrease the bias. This leads to a discussion on the initialisation strategy of the model configuration in case no eddy parametrisation is activated.

I recommend a major revision

**1 Major Comments:**

- At the end of the paper, I am still wondering if this paper is a paper analysing possible sources of deep bias in climate model using a hierarchy of climate model with various ocean resolution or if this paper is a description on possibility open by unstructured mesh ocean model for climate application with an overview of the improvement generated by the local refinement. In the first case, the paper is maybe not adapted for GMD. In the second case, the analysis is only focused on the deep bias and nothing else. So it is not enough to convince me it is worthwhile to use this capability in a climate model for decade to century. There is no evaluation of other basic climate index as sea ice, ACC, AMOC, meridional heat transport …

- Discussion about the contribution of Gibraltar need to be strengthen (more detailed on the geometrical issue, overflow representation and water masses properties at the Gibraltar sill)

- In your 5 experiments, one of them do not have the same atmospheric model. The vertical profile suggest the atmospheric model resolution could also lead to strong bias reduction. For clarity, you should focus only on those having the same atmospheric model.

- As you mentioned a link between the deep layer and the surface via the mixed layer, you should discuss in more details what could affect the mixed layer depth intensity and location (path of the North Atlantic current, surface fresh water flux, heat flux, restratification process via eddy activity …). About the overall idea of the initialisation strategy, I found it interesting. As it is included in the result section, I think you have to try it and show result on the initial bias in the HR case. You mentioned the GM eddy parametrisation is the key in LR to avoid 'overshoot' of the bias because it is fully active from the start in LR. Why not run in parallel to your idea on 3d T/S restoring during the spin up of the eddy fields (something like GM fully active from the start with a decreasing intensity over a specific time scale).

In Minor comments, I went through the manuscript from the beginning. Some comments are related to the one mentioned above.

**2 Minor Comments:**

**Abstract**

- P1L6: 'we find that two major sources at the surface are responsible for the deep bias in the deep Atlantic': Please briefly mention these 2 mechanisms.

**Introduction**

- P1L21: You mentioned a major biais is present in CMIP5. Could you add references to it in addition to your illustrations?
- P2L2: You should reformulate "…, as well as climate change (…) that is, errors are larger …" It is not easy to understand.
- P3L3: This is the first time in the main text you are using AWI-CM acronym, I think you should defined it here.
- P3L11 and elsewhere: Be careful when using 'eddy resolving' term. I am not convince you are, even in the location reddish in your figure 2. You should precise where you are eddy resolving or permitting. In introduction, I can suggest something like '… a strong case to aim for a high resolution (X km or higher) in eddy active region …'

**Model configuration**

- P3L18: just mention the acronym here as you explain it before (see comments above).
- About GM details, I am sure that how to define the location where you activate GM and how to make the transition from 'off' to 'fully active' trigger a lot of discussion in your group. My question is: should it be dependant of the Rossby Radius instead of prescribed resolution threshold (25km and 50km)? At 25 km a lot of eddy active region are still not eddy resolving. Could you explain more why you choose these numbers (25 and 50), what are the sensitivity of your ocean model to these numbers?
- You should specify also in your model configuration
  - Your input data for the bathymetry
  - Your vertical coordinate system and number of vertical level and resolution range
  - If you are using some icebergs representation, how do you represent iceberg (iceberg model or prescribed pattern, melt set in surface or spread between surface and iceberg draft depth) and how you compute its calving rate.
- P4L1 : try to avoid pages with figures, tables and with only a few lines of text at the bottom. It is really easy to miss these lines.
- About the XR resolution, you should just mention it in the conclusion as perspective and remove reference to it. In the main text, I found it not useful, as you do not show and discuss any result from this configuration.

**Results**

- P6L30: The figure 3 do not represent a drift. So please reformulate.
- About the S profile there is some differences which seems not related to resolution:
  - Surface salinity error are from -0.2 to 0.2 without clear resolution dependence. So as it is a couple run, if you change your atmospheric resolution (REF vs LR) or you oceanic resolution (LR, MR, MR0 and HR), your surface fresh water forcing can change. So, I am wondering if your surface fresh water forcing in all your run is

similar. As you discuss impact of mixed layer depth on error in depth, I think it is quite important for the discussion in section 3.4.

- o In depth (deeper than 1500m) the resolution of the atmospheric model seems to play a big role in it. All the model using T127 atmospheric model have the same error. It is less clear in temperature but it still looks significant deeper than 2000m.
- o You should add discussion on it or maybe remove REF simulation from the paper.
- P7L5: By stronger deep cell, what do you mean? do you mean deep overturning cell?
- All the discussion about Gibraltar:
  - o Could you add precision about the geometric error in your configuration (ie model strait width compare to reality)?
  - o In Figure 6, we clearly see that the salinity in depth is much more saline than the observations. What is the quality of the water masses going out of the Med. Sea at Gibraltar? Does it impact your analysis?
  - o Gibraltar is a shallow sill and the connection with the deep layer of the ocean is made via cascading of the dense water (Gibraltar overflow). However, the modelisation of this process is quite challenging in ocean model. So, is the Gibraltar overflow well represented in yours simulations? If no, what are the impact of it on your simulations and sensitivity. You should mention the Med. overflow in your discussion, its representation in FESOM and its importance compare to the geometrical factor you mentioned.
- About the discussion in surface conditions:
  - o See comments earlier on fwf
  - o P8L18: 'no heat sources': Could you precise if you are using a geothermal heating. If yes, maybe reformulate the first sentence.
  - o Could you mention the effect of the contribution of the advection from the other basin into your analysis domain.
  - o Gulf Stream and NWC: There is many modelling paper reporting issue in modelling these area, discussing the possible reason for it and the impact on the large scale. You should not only mention resolution as possible reason. You can mention for example the numerical scheme used (penduff et al., 2007: https://www.ocean-sci.net/3/509/2007/os-3-509-2007.pdf), or the representation of the DWBC (Zhang and Vallis, 2007: https://journals.ametsoc.org/doi/10.1175/JPO3102.1). Resolution dependence is also visible in Marzocchi et al., 2015: https://www.sciencedirect.com/science/article/pii/S0924796314002437#f0010)
  - o P10L2 please precise 'This region (hatched in Fig. 7). Do you mean the difficulty to simulate a correct NWC and GS ?
  - o You mention that the issue with the Gulf Stream and NWC is not in direct contact with the outcropping isopycnals you are interested in but the representation of the GS and NWC strongly impact the North Atlantic Current which reach the latitude you are interested in. So it could be the location of the outcropping region is determined by the path of the NAC. Could you add discussion about this.
- About the along-isopycnal bias propagation:
  - o See comments about Gibraltar above

- For the mixed layer source, see comments about surface fwf above. About the realism of the >500m convection, could you show comparison with observation or at least reference showing what the mixed layer depth should be.
- As you are talking about deep bias, I think is is worth adding discussion about the Nordic Sill overflow. Is the representation of the Nordic sill in your various configuration affect your conclusions?
- 4 supplementary documents in half a page of discussion I found it too much. Could you find a way to represent the point you want to make in a figure? Often reader like me do not take the time to get back on their browser, find the link, click on it and watch 4 movies.
- You focus on the large improvement between LR and HR, but I found that there is also a large improvement between REF and LR (it let suggest also that the atmospheric model resolution is also important in decreasing the bias in depth.). See comments above on maybe removing REF from the document as LR and REF has roughly the same resolution.
- Please reformulate the conclusion of this section based on the comments above.
- About the SST bias you mention at the end, please mention a reference to a figure.
- Displacement and tilt of isopycnal:
  - You explain why the slope of the isopycnal is different but I think you should add clearly, why this leads to temperature bias along the isopycnals?
  - All your paper is focussing onto the depth 1000m. So I suggest for clarity to remove the discussion on the 200-300m depth range P15L7 to L14.
  - In your supplementary materials we clearly see in the LR case an error propagating from the Good Hope cap toward south America. Do you know why this propagation and not a bias intensifying all along the Atlantic Southern ocean?
  - You mentioned that this strong bias in the Atlantic is due to difficulty of GM to balance the Ekman transport. So, why the error is so large in the Atlantic sector only? The other sector are quite good in LR and REF compare to HR.
- Initialisation method:
  - P15L17: I found the mention of 'usually based on a smoothed climatology as done in this study' confusing. I suggest to remove it. If you effectively smoothed the climatology, mention it in the previous sentenced and in the model configuration section.
  - P15L19: what is the time scale you imply exactly by 'fast' adjustment? days? months? years?
  - About the example you mention (bias in the east North America), I will be more cautious. I agree that if the Gulf Stream is to north, you will have a warm bias in the Northern Recirculation Gyre but based on the information you show, we don't know if PHC is representing this coastal area with strong boundary current correctly (you have strong temperature front in this area). You should at least put PHC sst in Figure 7.
  - Could you add precision about the time needed for the eddy fields to develop in your configuration?

- o P15L28: I think you should add a specific plot to show this instead of claiming it 'evident' on a supplementary material video. I had to watch back and forth frame by frame to be convince.
- Conclusions:
  - o In the model configuration section and introduction, you insist a lot on the local resolution, its benefice to run climate model. I was expecting it to be mentioned at the beginning and in a stronger way than you did.
  - o I found the word 'the three worst performing CMIP5 model' not well chosen here without mentioning the criterion used for the assessment.
  - o Rewrite the discussion on Gibraltar based on the comments on the overflow and Med. Sea water property.
  - o P17L33: By 'outcropping often happens too far to the north compared to observations', please clearly specify what you imply? Do you imply that isopycnals outcrop in a region with stronger heat fluxes, warmer atmosphere …?
  - o Most of your paper is on the deep bias and you mentioned an example of bias developing at 200m depth. As I mentioned earlier, to keep your paper focus you should maybe get rid of the paragraph discussing this.
  - o P18L5 to L13: You should move this paragraph earlier in the conclusion, maybe at the beginning.
  - o P18L29: 'we have shown major improvement'. You need to add limitation to this statement. You only show major improvement on the T/S bias at 1000m. We don't know at all if it improve the MOC, MHT, bottom water formation.

**3 Figure and table comments**

- Fig. 1:
  - o replace left/right by 'a)' and 'b)' and add it on the figure
  - o Comments on what you should see 'In the first hundreds meter …' should go into the text not in the caption.
  - o Mean abs. error in the top 300m is hard to see (overwritten by blue and red line), maybe consider using transparency and envelope.
- Fig. 2: remove XR if you follow my comments on removing XR from the text.
- Fig. 3:
  - o As for Fig. 1, replace left/right by 'a)' and 'b)' and add it on the figure
  - o Comments on what you should see 'With the medium- and …' should go into the text not in the caption.
- Fig. 6: split left column from the right column and put a label for each figure and use it in the caption.
- Fig. 7: add a label for each figure and add PHC sst figure and maybe use the same colorbar as in figure 6.
- Fig. 8: remove red line, as they are not commented on this figure.
- Fig. 9: If you remove discussion on 30.8 and 30.5 isopycnal line, do not forgot to remove it here. You are not commented the green line in this figure, so please remove it.
- Supplementary movies : please and a date on each frame, so we know where we are when we look at it (discussion on initial condition)
- Table 1: Add interannual std and climate change signal in the top or bottom cells.

- Table 2: Remove XR line and remove the internal name (not used in the manuscript).

---

## Author Comment (AC1) · 3 Jan 2019

We thank both reviewers for their constructive and thorough comments. We focused our efforts on two main points raised by the reviewers: First, we added an analysis of the top-of-the-atmosphere (TOA) balance and show that the need for TOA tuning is smaller for the medium-resolution grids, and even smaller for the high-resolution grid. Second, to further support the analysis, we computed "performance indices" (see definition below) for the different simulations that grade the quality of the overall simulated climate (Reichler and Kim, 2008). This shows that the higher resolution grids with reduced biases in the deep ocean do not result in a degradation of the overall simulated climate, but rather come along with an improvement.

Finally, since the intent of the paper is to mainly introduce AWI's CMIP6 model configurations in GMD and to document the identified sensitivities to spatial resolution, there is, of course, a certain level of compromise in terms of what is shown here, and detailed budget analyses need to be left for future oceanographic studies in other journals.

Reviewer comments are in blue, our response is in black.

**Anonymous Referee #1**

*Review of "Sensitivity of deep ocean biases to horizontal resolution in prototype CMIP6 simulations with AWI-CM1.0" by Rackow and Co-authors*

*The manuscript looks into the role of increased horizontal resolution in select regions of the ocean component of a coupled model in addressing, i.e., reducing, some of the deep ocean temperature and salinity biases in the Atlantic and Southern Ocean Basins. The authors argue that the ocean biases develop primarily from the surface, propagating along related isopycnals to the deep ocean. Higher horizontal resolution in the outcrop regions of these isopycnals appears to reduce such biases at depth. Although it is an interesting piece of work, I find the analysis rather superficial and qualitative, for example, relying on animations, rather than quantitative analysis.*

We thank the reviewer for the assessment and the constructive comments. The animations were done after the analysis was performed and were only meant as additional (supplementary) information, in order to better illustrate what is going on.

*I recommend major revisions along the following lines:*

*1. The Introduction actually introduces some physical mechanisms based on several previous studies concerning how deep temperature and salinity biases can emerge. In particular, the role of vertical mean and eddy heat transports is mentioned. Unfortunately, the manuscript does not get back to these points until the last section, and more importantly does not present a quantitative analysis exposing the role of various mechanisms. I strongly think that budget analyses should be included in the manuscript, in particular, exposing the changes in vertical eddy transports with increased horizontal resolution.*

In our prototype simulations, analyzed in this manuscript, we only saved monthly mean output. Therefore, it is impossible to carry out a thorough budget analysis suggested by the reviewer. However, the final CMIP6 simulations will most likely include much more (eddy) diagnostics and output at higher frequency, so that the issue will be revisited. Moreover, work has recently started to implement more 'online' eddy diagnostics directly into the code of FESOM1.4's successor "FESOM2", which will allow for detailed budget analyses in future simulations. At this stage, we can state that the presented results (reduced drift at depth around 1000m, smaller biases in the deep ocean) are consistent with findings previously published in the literature, and our along-isopycnal analysis adds to the existing discussion.

Since we agree that further analyses would certainly make sense, we decided to add a cautionary note to the summary:
*"Overall, we have shown major improvements when using medium-resolution (MR) and high-resolution (HR) meshes on representing the hydrography in the deep ocean. These grids are partly eddy-resolving and partly at most eddy-permitting, so that eddy parameterizations still need to be applied locally. This calls for dedicated in-depth analyses of eddy heat fluxes (and budgets) and their representation on multi-resolution unstructured grids in future studies."*

We also added the information about the available monthly-mean output for other readers:
*"In this study, we will analyze monthly-mean output of five pre-industrial simulations over a common 100-yr period."*

*2. The authors identify three regions for the source of deep biases. The first is the Strait of Gibraltar. I do not necessarily agree with the authors view that incorporation of tides will improve the representation of Mediterranean Outflow. The outflow / overflow processes require resolutions of order 10s of meters in the horizontal and meters in the vertical. Two possible solutions are an overflow parameterization and changes in the bottom / lateral topography at the outflow of the Strait of Gibraltar to minimize spurious entrainment.*

We agree with the reviewer that tides are most certainly not the panacea for the representation of the Mediterranean Outflow in climate models; there is, however, a role for the later spreading of waters from the Gulf of Cadiz into the North Atlantic (Izquierdo and Mikolajewiscz, 2018). Our intention was to list one possible remedy for some of the observed differences to the simulated climate. As suggested by the reviewer, we now also added a discussion of the other approaches that are discussed in the literature to the paper, with additional references to Wu, Danabasoglu, and Large (2007; overflow parameterization) and Izquierdo and Mikolajewicz (2018):

*"We hypothesize that at these resolutions, smaller issues become relatively more apparent, that is other processes might need to be included for a proper simulation of the Strait of Gibraltar outflow and spreading of Mediterranean Waters into the North Atlantic. Also, resolving the overflow processes at the Strait of Gibraltar would require resolutions on the order of tens of meters in the horizontal (Izquierdo and Mikolajewicz, 2018) and meters in the vertical direction, which is still far from the resolutions applied in this study.*

*Two possible solutions are therefore the use of an overflow parameterization (Wu et al, 2007), which is currently not implemented in the model, or systematic changes to the bottom (and lateral) topography at the outflow of the Strait of Gibraltar to minimize spurious entrainment. In order to simulate the correct spreading of Mediterranean Waters from the Gulf of Cadiz into the North Atlantic, another approach could be to add additional physics like the effect of tides (Izquierdo et al, 2016), which are usually not included in current climate models. Without tides, ocean models often simulate erroneous south-westward spreading, leading to stronger biases when compared to climatology than in simulations with active tides (Izquierdo and Mikolajewicz, 2018)."*

In the outlook, we now also mention an outflow parameterization as a possible step towards an improved representation, and that much more systematic efforts are needed to improve the representation of Gibraltar for different resolutions. In this study, the representation (width and depth) changed with increasing spatial resolution, see the plot below in an answer to reviewer #2. We did not try to keep the same geometry in Gibraltar for all the different meshes. Such future work is a necessary next step, and it is possible for us because the ocean model supports variable-resolution grids.

*The second source is identified as the erroneous downwelling associated with anomalously deep mixed layers in the northeastern North Atlantic. This statement is not justified. How do you know that the downwelling is erroneous and that the mixed layer depths are anomalously deep?*

This statement is based on previous findings from stand-alone ocean simulations within the coordinated 'CORE2' intercomparison project (Danabasoglu et al., 2014; their Figure 13), where it was shown that the mixed layer (using the "LR" grid) is anomalously deep in this area. For convenience, in the following we show the relevant part of their figure here (left: World Ocean Atlas; right: FESOM stand-alone with LR grid):

[Figure]

*(Figure excerpt from Danabasoglu et al., 2014;*
*https://www.sciencedirect.com/science/article/pii/S1463500313001868?via%3Dihub )*

The MLD pattern (for MLD>500m, see green contours in supplemental animation S1 of LR) reproduces the behaviour known from the uncoupled simulation. Since the Gulf Stream in

the REF and LR simulations is too zonal and reaches the northeastern North Atlantic, part of the flow has to downwell here (negative w's), which we suspect could explain part of this deficiency by entraining waters and deepening the mixed layer. We have added this information to the text:

*"Since the Gulf Stream in the LR (and REF) simulations is too zonal and reaches the northeastern North Atlantic, part of the flow has to downwell here, which we suspect could explain part of this deficiency by entraining waters and deepening the mixed layer. Other factors influencing the mixed layer depth could be biased buoyancy fluxes or the restratification process via eddy activity."*

*The third source is presumably related to a displacement of isopycnals which are identified as too steep when eddies are parameterized. First, the analysis is not quantitative and I do not really follow the argument.*

It is a classical hen-and-egg problem where you can not decide what comes first, the steepened isopycnals or the temperature biases?
The argument is that a strong warm (cold) bias dipole like in Fig.10b,left) will decrease (increase) density in the 600-1000m range. The 31.8 isopycnal (black) will therefore be lower on the northern side and higher on the southern side of this bias dipole when compared to climatology (magenta line), steepening the isopycnal. The other way round, isopycnal slope is partially controlled by eddies (or by the eddy parameterization), which tend to flatten isopycnals. If the slope diverges strongly from climatology --be it because eddies are not sufficiently resolved or because the GM coefficient is locally too small-- this should lead to similar temperature biases. Fig.10b,right) shows how a more realistic isopycnal slope in MR coincides with much smaller temperature biases. We have some control on the isopycnal slope by locally resolving eddies or, potentially, by locally tuning the GM coefficient in future simulations:
*"Since we were are using a default GM coefficient for all simulations, it can be argued that a regional tuning of GM with a horizontally varying coefficient (Visbeck et al, 1997; Danabasoglu et al, 2012) could lead to a better simulation of the Southern Ocean in low-resolution AWI-CM configurations."*

In MR, the isopycnal slope of the 31.8 contour between 40°S and 45°S at 10.5°E is close to climatology (about 300m per 5° of latitude), and it is more than doubled in LR (about 700m per 5° of latitude). We added this information to the text:
*"Already at medium resolution (MR), the simulated isopycnal slope is about halved compared to LR and much closer to the observed slope (Fig.10b, right)".*

*Second, this is likely due to the issues with the details of the mesoscale eddy parameterization used. A description of the parameterization as implemented in the model should be included. Furthermore, since the REF case is much cheaper, a couple of cases with modified versions of the parameterization could be tested as alluded to in the text.*

We added a basic description of the mesoscale eddy parameterization and refer for further details to Wang et al (2014):

*"In order to parameterize eddies at non-eddy resolving resolutions, the Gent and McWilliams (1990) parameterization (GM) is applied with isoneutral diffusion (Redi, 1982). All prototype simulations used a reference diffusivity $K_{ref}(x,y) = 600\ m^2\ s^{-1}$, which is scaled by the local resolution (Wang et al., 2014), and a GM coefficient $K_{GM}=K_{ref}/2$. As detailed by Wang et al (2014), tapering functions following Danabasoglu and McWilliams (1995) and Large et al (1997) are also applied to $K_{GM}$.*
*Depending on the local resolution, the GM parameterization in FESOM1.4 is smoothly switched off at resolutions smaller than 25 km (red areas in Fig.2), and its effect increases linearly until 50 km, when the parameterization is fully active (Wang et al., 2014). For example, the parameterization is locally switched off when using the 'MR' and 'HR' meshes, which are locally eddy-resolving, and it is generally active in the lower-resolution 'LR' mesh (see next sections). The thresholds of 25 km and 50 km can be considered to be tuning parameters and were chosen in stand-alone simulations with FESOM1.4 using the LR grid. For the Arctic, changing the numbers can result in too diffuse boundary currents (Wang et al., 2014) and their (automatic) choice remains an important research topic for multi-resolution climate applications with AWI-CM."*

There is certainly scope for an improved eddy parameterization/implementation in our model. The possible modified versions of the eddy parameterization referred to in the text (e.g. Visbeck et al) are not yet implemented in the model and will most probably be tackled in future projects.
However, as a future perspective, we strengthened the point in the discussion that by locally tuning the coefficient (e.g. by using high-res simulations as a template) we could possibly get similar answers with a low-resolution model:

*"Since we were using a default GM coefficient for all simulations, it can be argued that a regional tuning of GM with a horizontally varying coefficient (Visbeck et al. 1997, Danabasoglu et al. 2012) could lead to a better simulation of the Southern Ocean in low-resolution AWI-CM configurations. Moreover, high-resolution simulations and their effective $K_{GM}$ could also serve as a template for the regional tuning of low-resolution simulations."*

*Incidentally, I am not sure what is meant by mean absolute error. Is this the root-mean-square (rms) error?*

The absolute error is computed at every gridpoint as the absolute difference |T_m - T_o|, where T_o is the observed and T_m is the modeled value (e.g. potential temperature). In the end, a horizontal mean over all gridpoints is performed to get the "mean absolute error". This way, smaller values are indicative of true improvements and are not caused by compensating biases of different sign. We added an explanation of the term to the caption of Table 1.

*3. The text refers to higher resolution configurations as (regionally) eddy-resolving in various places. Are they? As far as I can tell, they are still mostly eddy-permitting. A definition of what is meant by eddy-resolving and spatial maps of eddy-permitting and eddy-resolving regions for each configuration should be included. The text says "resolving the Rossby radius", but that is not a quantitative statement.*

The map for the HR grid is given in the paper by Sein et al. (2016), their Fig.4c (see figure below).

[Figure]

*(**Fig.4c in Sein et al., 2016**: Designing variable ocean model resolution based on the observed ocean variability) **Green areas** are eddy-resolving, e.g. over the Western Boundary Currents, **yellow areas** are eddy-permitting, e.g. in the ACC, and **red areas** need to fully parameterize the effect of eddies.*

We added this information to the text:
*"For a spatial map of the eddy-permitting and eddy-resolving regions on the HR grid, please refer to Fig.4c in Sein et al. (2016)."*

Following the study by Hallberg (2013), the transition between eddy-permitting and eddy-resolving grids is two grid intervals per Rossby radius, but finer resolution might still be needed to capture mesoscale eddy dynamics. We rewrote the paragraph as follows:

*"Ultimately, we will target coupled configurations with a globally eddy-resolving mesh, which implies "resolving the Rossby radius" almost everywhere with at least 2 grid intervals per Rossby radius (Hallberg, 2013). Using this criterion, we have recently reported on the development of such a 'frontier' mesh (XR; see Fig.2, right*

*globe), with resolution capped at 4km (7km) in the Arctic (Antarctic) (Sein et al., 2017). Sein et al. (2017) note that an even finer resolution will be required locally to fully capture mesoscale eddies."*

*What is the physical justification for cutting of the eddy parameterization below 25 km resolution, knowing that the resolutions are mostly on the eddy-permitting side?*

Please refer to the answer to reviewer #2 below. Red areas in Fig. 2 show where the GM parameterization is switched off. We added this sentence to the figure's caption.

*Also, as far as I can tell, the number of vertical levels is not given in the manuscript.*

The number of layers is 46 for all meshes. The levels are located at 0, 10, 20, 30, 40, 50, 60, 70, 80, 90, 100, 115, 135, 160, 190, 230, 280, 340, 410, 490, 580, 680, 790, 910, 1040, 1180, 1330, 1500, 1700, 1920, 2150, 2400, 2650, 2900, 3150, 3400, 3650, 3900, 4150, 4400, 4650, 4900, 5150, 5400, 5650, 5900m. We added this information to the text.

*4. I am unsure if all the cases represent an apples-to-apples comparison. Specifically, these are fully coupled, pre-industrial simulations. Changes in one component will undoubtedly introduce the need to retune the top-of-the-atmosphere (TOA) radiation budget. Please provide a table with the TOA values for each configuration. My point is that if the reduced bias cases show large negative TOAs in comparison to the REF case, then when the coupled model is retuned, then it is possible that the deep ocean biases will reappear.*

We think it is a fair comparison since we use the same atmospheric settings for all (T127) simulations, except that we changed the ocean grid. The atmospheric parameters and settings were not tuned to a particular AWI-CM configuration; instead, they reflect typical tuned values of the sister model MPI-ESM, which uses the same atmospheric component (ECHAM6) but a different ocean model (MPIOM). We deliberately decided not to tune the TOA for the prototype simulations analyzed in our manuscript because it was our intention to isolate the sensitivity to the ocean resolution first.

As correctly pointed out by the reviewer (and stated in the last paragraph of our revised manuscript) any change to the model components will make a retuning of the TOA necessary, and this is done for the final CMIP6 control runs.

We want to emphasize again that we did not aim for a balanced TOA close to zero already after 100 years of simulation. For example, the deep ocean in a (present-day) simulation with the REF grid slowly drifts over 1500 years before it reaches a quasi-equilibrium (Sidorenko et al. (2015), Rackow et al. (2016)), and similar drifts have been reported for pre-industrial simulations. Tuning the TOA to zero at the beginning could potentially produce unwanted effects in the balanced state. The higher-resolution ocean grids in the pre-industrial configurations analyzed here appear to show much reduced drifts compared to REF (Hovmoeller plots in Fig. 4).

The global net balance at the TOA in the last 30 years of the simulations with T127 atmosphere is as follows:

LR:  1.02 W/m$^2$ (low-resolution)

MR:  0.89 W/m$^2$
MR0: 0.62 W/m$^2$ (medium resolutions)

HR:  0.44 W/m$^2$ (high resolution)

This indicates that the need for TOA tuning is smaller for the medium-resolution grids, and even smaller for the high-resolution grid.

The REF simulation (with a different atmosphere at coarser T63 resolution), which has the strongest deep ocean bias at 1000m, has a better TOA balance (0.58 W/m$^2$) than the LR and MR/0 simulations: it is more on the level of the medium- to high-resolution configurations with T127 atmosphere. Nevertheless, it has the strongest deep ocean bias, similar in magnitude to LR with a TOA balance of 1.02 W/m$^2$ (Fig.3a). This already indicates that the (global) TOA balance is not directly related the North Atlantic deep ocean biases, which are determined by localized phenomena (performance over outcropping regions, Strait of Gibraltar). Thus, a global TOA retuning is unlikely to affect the deep ocean bias significantly. We therefore added a cautious note to the text that a retuning might impact the deep ocean biases:

*"Tuning could potentially affect the deep ocean simulation, although the global TOA balance in particular appears not to be directly related to the magnitude of North Atlantic deep ocean biases (not shown)."*

*Additionally, please include comparisons of the Atlantic meridional overturning circulation (AMOC), Labrador Sea Deep Water formation / mixed layer depth, and the northward heat and salt transports to show that the reductions in the deep biases are not occurring at the expense of degradations in several other climatically important fields.*

We agree that, especially in fully coupled systems, changes or improvements in one component could potentially negatively affect other climatically relevant processes or fields. Since there is a vast number of important diagnostics (which were already discussed in the introductory papers by Sidorenko et al. (2015) and Rackow et al. (2016)) and in order to give a comprehensive answer, we extended the atmospheric "Performance Index" (PI) analysis, which was originally introduced by Reichler and Kim (2008), to ocean fields. How these indices are computed is detailed in the new Appendix B.
It basically summarizes the modelled mean climate into a score that can be quantitatively compared to other configurations and models. This modified version of the PI for important atmospheric parameters was already used in Sidorenko et al. (2015) and Rackow et al. (2016) to judge the modelled climate of AWI-CM; here we introduced the oceanic PIs for the first time.

The *atmospheric* PI for our configurations are as follows (<1 means better than the average of the considered CMIP5 models):

REF: 1.03
LR: 0.87
MR: 0.81
MR0: 0.79
HR: 0.80

The improvements, thus, roughly follow the oceanic resolution, so that bias reductions in the surface and deep ocean are not occuring at the expense of degrading atmospheric fields. Instead, the atmospheric simulation appears to improve, especially when going from T63 to T127 (REF->LR). However, the simulation of the atmosphere further improves when going from LR to the higher ocean resolutions (all with T127).

The oceanic PI are as follows:

|      | Global | North Atlantic | Southern Ocean |
|------|--------|----------------|----------------|
| REF  | 0.87   | 0.98           | 0.68           |
| LR   | 0.72   | 0.80           | 0.74           |
| MR   | 0.64   | 0.62           | 0.62           |
| MR0  | 0.61   | 0.57           | 0.62           |
| HR   | 0.66   | 0.63           | 0.62           |

Again, while going from REF (with T63) to LR (with T127) generally improves the ocean simulation, except in the Southern Ocean-, going to even higher ocean resolutions while keeping the same atmosphere (T127) further improves the simulation.

We conclude that the bias reductions in the deep ocean do not come at the expense of degradations in the simulation of the whole system.

*5. In the last paragraph of section 3.4, it is stated that "higher spatial resolution is needed . . . . to better simulate the position of the Gulf Stream." I thought that there were studies in literature showing that the high resolution is not really the silver bullet. Perhaps an expanded discussion should be included here.*

We agree that we did not give a balanced discussion in the manuscript and therefore extended the discussion of the "Gulf Stream separation" topic. As was suggested by reviewer #2, we expanded the discussion as follows:

*"While a strong resolution-dependence was also shown by Marzocchi et al. (2015), there are additional ways for getting a more realistic Gulf Stream separation. These include details of the numerical scheme that can affect current-topography interactions (Penduff et al. 2007) or the representation of non-local dynamics that impact the formation of a northern recirculation gyre along the North American coast, such as the Deep Western Boundary Current downstream of Cape Hatteras (Zhang and Vallis, 2007) and the cold Labrador Current northward of the Gulf Stream front (Sein et al., 2017)."*

*Also, I do not really follow the argument made in the last paragraph of section 3.4.1.*

What we want to say here is that these biases are likely due to a northward shift of the surface isopycnals, indicating a shift of the water masses in this area; a southward shift (by flattening the isopycnal slope) could thus result in strongly reduced biases, as indeed seen in the MR simulation. We rewrote the paragraph as follows:

*"Interestingly, the surface representation (SST bias) of this warm/cold interior bias to the west of Cape Agulhas and a similar dipole-like bias in the Brazil-Malvinas Confluence region are cleanly separated into their warm and cold parts by the $\sigma_1$=30.5 isopycnal surface contour (red contour in Fig.10a, left) in LR. This suggests that these biases could be caused by shifted water masses as indicated by the erroneous northward shift of the $\sigma_1$=30.5 contour, leading to a warm bias on its northern side and to a cold bias on its southern side. Flattening the slope would result in a southward shift with potentially reduced biases. Indeed, the surface biases are strongly diminished in MR (Fig.10a) with better resolved eddies and the associated flatter isopycnals, which are a close fit to the target contours from PHC (Fig.10b)."*

**Anonymous Referee #2**

*"Sensitivity of deep ocean biases to horizontal resolution in prototype CMIP6 simulations with AWI-CM1.0" by Thomas Rackow et al.*

*Rackow et al. are describing a hierarchy of climate model using the AWI-CM. They present the capability of the ocean model on unstructured mesh for climate application. The focus is on the benefit of using local refinement in eddy active region to decrease the deep temperature bias. In addition to that, they discuss why the high resolution decrease the bias. This leads to a discussion on the initialisation strategy of the model configuration in case no eddy parametrisation is activated. I recommend a major revision*

We thank the reviewer for a thorough review and the detailed suggestions. We implemented the most pressing ones and mostly followed the suggestions for new figures, for the layout of figures and movies, and for restructuring of the text. Furthermore, the "performance index analysis" (see new Appendix B) shows that the medium- and high-resolution configurations do not degrade the overall climate (both ocean and atmosphere), but rather improve the whole simulation.

**1 Major Comments:**

*- At the end of the paper, I am still wondering if this paper is a paper analysing possible sources of deep bias in climate model using a hierarchy of climate model with various ocean resolution or if this paper is a description on possibility open by unstructured mesh ocean model for climate application with an overview of the improvement generated by the local refinement. In the first case, the paper is maybe not adapted for GMD. In the second case, the analysis is only focused on the deep bias and nothing else. So it is not enough to convince me it is worthwhile to use this capability in a climate model for decade to century. There is no evaluation of other basic climate index as sea ice, ACC, AMOC, meridional heat transport …*

To provide a comprehensive picture of the simulation quality, we added "performance indices" for both the ocean and atmosphere (see also answer to reviewer #1 and the new Appendix B). This is the first time that performance indices have been computed for an ocean model and the scores clearly highlight the improvements in the higher-resolution ocean configurations.

AWI-CM REF (with T63 atmosphere) has been extensively evaluated in two papers in Climate Dynamics (Rackow et al. 2016, Sidorenko et al. 2015), discussing all above mentioned indices and other relevant fields. While already showing that AWI-CM, in its reference configuration, is comparable to other models, the papers also highlighted the deep ocean bias as one of the most prominent issues requiring work.
In the present manuscript, which aims at improving upon this situation before our group will participate in CMIP6, we put this bias into a more general context (CMIP models show a bias like this as well) and propose a remedy via locally increased ocean resolution. We indeed focus here mainly on the deep ocean bias, but participating in CMIP6 -with model evaluation performed by the whole community- could show that this new technology is generally promising with respect to other parameters as well.

*- Discussion about the contribution of Gibraltar need to be strengthen (more detailed on the geometrical issue, overflow representation and water masses properties at the Gibraltar sill)*

We added a clarification on what we initially referred to as the "geometrical issue" and added a new Figure 9, which shows the spatial discretization of the Strait of Gibraltar for all the different meshes along with the local ocean depth. We also state in the revised manuscript that more systematic efforts are needed to properly tune the width and depth in low-resolution configurations. As suggested by reviewer #1, we also added references to Wu, Danabasoglu, and Large (2007; overflow parameterization) and Izquierdo and Mikolajewicz (2018; benefit of tides) as additional perspectives for further improvements.

[Figure]

**Fig.9:** *Spatial discretization of the Strait of Gibraltar in the 5 different model grids. The thick black line shows the true coastline as implemented in the Basemap plotting toolbox, using data from GSHHS ([http://www.soest.hawaii.edu/wessel/gshhs/gshhs.html](http://www.soest.hawaii.edu/wessel/gshhs/gshhs.html)). Triangular elements are shown with thin black lines, colors depict the local ocean depth in meters.*

Usually the bathymetry undergoes one or two steps of grid scale smoothing to make the model run stable (Wang et al., 2014). The effect of this smoothing is less evident for the higher resolution grids, since smoothing takes places on much smaller scales and therefore the resulting bathymetry is closer to the "observed" input bathymetry data. While the increased ocean resolution is expected to decrease spurious numerical mixing, the improved representation of the bathymetry to the West of the Strait of Gibraltar is expected to further add to the realism of the model and the simulated plume. However, we don't use an overflow parameterization in the current AWI-CM model yet; this point was made clear in the revised manuscript.

*-  In your 5 experiments, one of them do not have the same atmospheric model. The vertical profile suggest the atmospheric model resolution could also lead to strong bias reduction. For clarity, you should focus only on those having the same atmospheric model.*

The reviewer is certainly right that the step from "REF" to "LR" includes simultaneous changes of both the ocean grid as well as the change from a T63 to T127 atmosphere, which was not made sufficiently clear before. We clarified this point at several places in the revised manuscript and modified the discussion accordingly. To facilitate better comparability with previous studies, where REF/T63 was introduced as a benchmark configuration of AWI-CM against which future configurations should be compared, we still decided to keep the REF configuration in the revised manuscript.

*-  As you mentioned a link between the deep layer and the surface via the mixed layer, you should discuss in more details what could affect the mixed layer depth intensity and location (path of the North Atlantic current, surface fresh water flux, heat flux, restratification process via eddy activity ...).*

We added a discussion of these points to the text and refer the reader to the discussion and Fig.13 of Danabasoglu et al (2014) (see the relevant part of their figure above in an answer to reviewer #1):

*"This is a feature that has already been identified in uncoupled FESOM simulations using the LR grid as part of the CORE-II intercomparison project (Danabasoglu et al, 2014; their Fig.13). Since the Gulf Stream in the LR (and REF) simulations is too zonal and reaches the northeastern North Atlantic, part of the flow has to downwell here, which we suspect could explain part of this deficiency by entraining waters and deepening the mixed layer. Other factors influencing the mixed layer depth could be biased buoyancy fluxes or the restratification process via eddy activity."*

*About the overall idea of the initialisation strategy, I found it interesting. As it is included in the result section, I think you have to try it and show result on the initial bias in the HR case. You mentioned the GM eddy parametrisation is the key in LR to avoid 'overshoot' of the bias because it is fully active from the start in LR. Why not run in parallel to your idea on 3d T/S restoring during the spin up of the eddy fields (something like GM fully active from the start with a decreasing intensity over a specific time scale).*

We thank the reviewer for the positive feedback. This is an immediate idea based on our results, and that's why we decided to mention this idea directly at the end of the results section in the submitted manuscript. This is beyond the scope of this paper, but we are glad that the reviewer agrees with the relevance of testing different initialisation techniques and their potential for improving climate simulations. Since the 3D T/S restoring and other techniques still need to be implemented and tested carefully, they need to be left for future studies.

The caveat with using GM fully active from the start in high-resolution simulations, but with a decreasing intensity over the timescale when the explicitly resolved eddy field is developing, is that - in our experience - GM tends to damp a significant part of the *resolved* eddies because the parameterization decreases baroclinicity. However, we agree that this is a plausible conjecture that would be worth testing in the future.

*In Minor comments, I went through the manuscript from the beginning. Some comments are related to the one mentioned above.*

**2 *Minor Comments:**

*Abstract*
*- P1L6: 'we find that two major sources at the surface are responsible for the deep bias in the deep Atlantic': Please briefly mention these 2 mechanisms.*

We extended the sentence in the abstract accordingly:

*"We find that two key regions at the surface are responsible for the development of the deep bias in the Atlantic Ocean, the north-eastern North Atlantic and the region adjacent to the Strait of Gibraltar."*

*Introduction*
*- P1L21: You mentioned a major bias is present in CMIP5. Could you add references to it in addition to your illustrations?*

Although being substantial in magnitude, to our surprise the deep ocean biases in the CMIP5 models did not receive a lot of attention yet. There is a paper for one specific model (EC-Earth) by Sterl et al. (2012), where maps of temperature and salinity biases in 1000m are shown. We are not aware of other papers discussing the CMIP5 bias specifically, so we only added the Sterl et al. (2012) paper:
*"A major bias present in CMIP5 models is reflected by a too warm and saline deep ocean compared to observations (e.g. in the EC-Earth model; Sterl et al. 2012)."*

*- P2L2: You should reformulate "..., as well as climate change (...) that is, errors are larger ..." It is not easy to understand.*

We rephrased the sentence as follows:
*"Importantly, the mean absolute error in deeper ocean layers is larger than the interannual variability (the standard deviation of annual means). It is also larger than the climate change signal as determined from RCP8.5 and RCP4.5 emission scenarios. Or formulated differently, deep ocean biases are larger than the signals we aim to predict, which may be cause for concern in non-linear systems."*

*- P3L3: This is the first time in the main text you are using AWI-CM acronym, I think you should defined it here.*

Done. We still mention in the model configuration section that AWI-CM was previously named "ECHAM6-FESOM", with citations of Sidorenko et al. (2015) and Rackow et al. (2016).

*- P3L11 and elsewhere: Be careful when using 'eddy resolving' term. I am not convince you are, even in the location reddish in your figure 2. You should precise where you are eddy resolving or permitting. In introduction, I can suggest something like '... a strong case to aim for a high resolution (X km or higher) in eddy active region ...'*

For the HR grid (10--60 km resolution), we now refer the reader to a map in Sein et al. (2016) where the eddy-resolving and eddy-permitting regions are shown. The MR grid, used for the standard CMIP6 cases, uses even higher resolution locally over the Gulf Stream area (Fig.2). As suggested by the reviewer, we therefore changed the text as follows:
*"[...] a strong case to aim for a high resolution (10 km and higher) in eddy-active regions not only in HighResMIP (Haarsma et al., 2016), but already for AWI's CMIP6 standard configuration."*

*Model configuration*
*- P3L18: just mention the acronym here as you explain it before (see comments above)*

Done. We now mention the acronym here (and the previous name ECHAM6-FESOM; see comment above).

*- About GM details, I am sure that how to define the location where you activate GM and how to make the transition from 'off' to 'fully active' trigger a lot of discussion in your group. My question is: should it be dependant of the Rossby Radius instead of prescribed resolution threshold (25km and 50km)? At 25 km a lot of eddy active region are still not eddy resolving. Could you explain more why you choose these numbers (25 and 50), what are the sensitivity of your ocean model to these numbers?*

At mid-latitudes, the Rossby radius is between 25 and 50km, which is why this initial simple choice was made. Still, 25km and 50km can be considered to be tuning parameters and where chosen in stand-alone simulations with FESOM1.4 (Wang et al., 2014), and those numbers worked well in practice and gave the best results (using the LR grid). For the Arctic, regarding sensitivities, changing the numbers can result in too diffuse boundary currents (Wang et al., 2014).

We agree that the criterion should be Rossby radius-dependent, and it already is in FESOM1.4's successor FESOM2. However, even in the formulation based on the Rossby radius, tuning is required because the scales of baroclinic instability do not only depend on the Rossby radius, but also on details of the potential vorticity gradient profile and the surface buoyancy gradient.

In summary, these numbers are currently considered to be tuning parameters and how to switch between "on" and "off" GM regions will remain an important research topic for multi-resolution applications with FESOM (Wang et al., 2014).

We added the following sentences to the text:
*"At mid-latitudes, the Rossby radius is between 25 and 50 km, which is why this simple choice was made. Still, the thresholds of 25km and 50km can be considered to be tuning parameters and were chosen in stand-alone simulations with FESOM1.4 using the LR grid. For the Arctic, changing the numbers can result in too diffuse boundary currents (Wang et al, 2014) and their (automatic) choice remains an important research topic for multi-resolution climate applications."*

*"Depending on the local resolution, the parameterization is smoothly switched off at resolutions smaller than 25 km (red areas in Fig.2)"*

*We added "The GM parameterization is switched off within the red areas." to Fig. 2's caption.*

*- You should specify also in your model configuration*
*o Your input data for the bathymetry*

The bathymetry in the model is based on a blend of several bottom topography data sets, as detailed in Wang et al (2014): North of 69°N, the International Bathymetric Chart of the Arctic Oceans (IBCAO version 2, Jakobsson et al., 2008) is used (2km resolution). South of 64°N, the General Bathymetric Chart of the Oceans (GEBCO) is used (1min resolution). The topography is computed as a linear combination of the two charts between 64°N and 69°N.

We added:
*"The bathymetry in the different grids is based on a blend of the IBCAO (Jacobsson et al, 2008) and GEBCO (...) bottom topography data sets, as detailed by Wang et al. (2014)."*

*o Your vertical coordinate system and number of vertical level and resolution range*

Done. We added information on the vertical levels to the text, both the number (46) and the exact levels.

*o If you are using some icebergs representation, how do you represent iceberg (iceberg model or prescribed pattern, melt set in surface or spread between surface and iceberg draft depth) and how you compute its calving rate.*

The model did neither use interactive icebergs nor spatio-temporal templates representative of iceberg drift for the distribution of land ice over the ocean. The runoff and calving from land was put into the ocean in a narrow band around Greenland and Antarctica, as is the default practice in the model.

*- P4L1 : try to avoid pages with figures, tables and with only a few lines of text at the bottom. It is really easy to miss these lines.*

Done. We removed single lines of text below figures 4 and 5 and below Table 2.

*- About the XR resolution, you should just mention it in the conclusion as perspective and remove reference to it. In the main text, I found it not useful, as you do not show and discuss any result from this configuration.*

The motivation for the medium-resolution grids is that we can run simulations with spatial resolution smaller than 10km in eddy-active regions (MR0/MR/HR) already today, even though we cannot easily run the global XR resolution over climate-relevant timescales yet. As XR is the future goal and constitutes the motivation for the design of our current medium-resolution meshes, we'd like to keep it in the main text.

*Results*
*- P6L30: The figure 3 do not represent a drift. So please reformulate.*

Done. We reformulated the sentence as follows:

*"Temperature and salinity show major improvements for medium- and high-resolution configurations, as seen from horizontally averaged temperature and salinity profiles [...]"*

*- About the S profile there is some differences which seems not related to resolution:*
*o Surface salinity error are from- 0.2 to 0.2 without clear resolution dependence.So as it is a couple run, if you change your atmospheric resolution (REF vs LR) or you oceanic resolution (LR, MR, MR0 and HR), your surface fresh water forcing can change. So, I am wondering if your surface fresh water forcing in all your run is*
*similar. As you discuss impact of mixed layer depth on error in depth, I think it is*
*quite important for the discussion in section 3.4.*

We have checked the net freshwater flux into the North Atlantic ocean (years 71-100) and we could not identify obvious strong differences. As an example, here is the freshwater flux for LR (approx. -0.2 surface salinity error) and HR (approx. +0.2 surface salinity error):

[Figure]

*o In depth (deeper than 1500m) the resolution of the atmospheric model seems to play a big role in it. All the model using T127 atmospheric model have the same error. It is less clear in temperature but it still looks significant deeper than 2000m. You should add discussion on it or maybe remove REF simulation from the paper.*

We added the following paragraph to the manuscript:
*"The simultaneous change of the ocean and atmospheric resolution from REF/T63 to LR/T127 leads to a clear improvement of the salinity profiles below 1500m, and all configurations with T127 atmosphere (LR, MR0, MR, and HR) share a very similar salinity bias in this range. While it is difficult to say what the relative influence between the atmospheric resolution change (T63 vs T127) and the switch of the ocean grid (REF vs LR) is, it appears that surface conditions can significantly impact deep ocean biases."*

*- P7L5: By stronger deep cell, what do you mean? do you mean deep overturning cell?*

Yes, "deep cell" was short for "deep overturning cell". We corrected this in the revised document.

*- All the discussion about Gibraltar:*

*o Could you add precision about the geometric error in your configuration (i.e. model strait width compare to reality)?*

We added a scale bar to Figure 9 to see the model strait width (in reality, it is about 15km).

*o In Figure 6, we clearly see that the salinity in depth is much more saline than the observations. What is the quality of the water masses going out of the Med. Sea at Gibraltar? Does it impact your analysis?*
*o Gibraltar is a shallow sill and the connection with the deep layer of the ocean is made via cascading of the dense water (Gibraltar overflow). However, the modelisation of this process is quite challenging in ocean model. So, is the Gibraltar overflow well represented in yours simulations? If no, what are the impact of it on your simulations and sensitivity. You should mention the Med. overflow in your discussion, its representation in FESOM and its importance compare to the geometrical factor you mentioned.*

For further discussion about the Strait of Gibraltar, please see our answer to the reviewer's major comment above (and answer to reviewer #1). We do not use an overflow parameterisation yet, this information was also added to the text.

*- About the discussion in surface conditions:*
*o See comments earlier on fwf*
*o P8L18: 'no heat sources': Could you precise if you are using a geothermal heating. If yes, maybe reformulate the first sentence.*

We are not using geothermal heating in our simulations. We added the sentence "*The ocean model also does not apply geothermal heating as lower boundary condition (e.g., Adcroft et al. 2001; Downes et al. 2016).*"

*o Could you mention the effect of the contribution of the advection from the other basin into your analysis domain. Gulf Stream and NWC: There is many modelling paper reporting issue in modelling these area, discussing the possible reason for it and the impact on the large scale. You should not only mention resolution as possible reason. You can mention for example the numerical scheme used (penduff et al., 2007: https://www.ocean-sci.net/3/509/2007/os-3-509-2007.pdf), or the representation of the DWBC (Zhang and Vallis, 2007: https://journals.ametsoc.org/doi/10.1175/JPO3102.1). Resolution dependence is also visible in Marzocchi et al., 2015:https://www.sciencedirect.com/science/article/pii/S0924796314002437#f0010)*

We thank the reviewer for pointing us to these papers. We added all the suggested references to the text. We now also explicitly mention the representation of important non-local dynamics (DWBC downstream of Cape Hatteras, impacting the formation of a cyclonic northern recirculation gyre, Zhang and Vallis 2007; cold Labrador current that meets the warm Gulf Stream, Sein et al. 2017):

"*While a strong resolution-dependence was also shown by Marzocchi et al. (2015), there are additional ways for getting a more realistic Gulf Stream separation. These include details of*

*the numerical scheme that can affect current-topography interactions (Penduff et al., 2007) or the representation of non-local dynamics that impact the formation of a northern recirculation gyre along the North American coast, such as the Deep Western Boundary Current downstream of Cape Hatteras (Zhang and Vallis, 2007) and the cold Labrador Current northward of the Gulf Stream front (Sein et al., 2017).”*

*o P10L2 please precise 'This region (hatched in Fig. 7). Do you mean the difficulty to simulate a correct NWC and GS ?*

We replaced "This region..." with "The cold temperature spot…". Yes, the cold spot is impacted by the difficulties to simulate a correct North West Corner and Gulf Stream.

*o You mention that the issue with the Gulf Stream and NWC is not in direct contact with the outcropping isopycnals you are interested in but the representation of the GS and NWC strongly impact the North Atlantic Current which reach the latitude you are interested in. So it could be the location of the outcropping region is determined by the path of the NAC. Could you add discussion about this.*

We extended the paragraph as follows:
*"Although the Gulf Stream and its extension could impact the location of the outcropping regions, the strong cold temperature spot (hatched in Fig.7b--d) is, however, not in direct contact with the deep ocean around 600--1000m depth via outcropping isopycnals and thus does not limit the analysis of the present manuscript, which is focused on the deep ocean."*

*- About the along-isopycnal bias propagation:*
*o See comments about Gibraltar above*
*o For the mixed layer source, see comments about surface fwf above. About the realism of the >500m convection, could you show comparison with observation or at least reference showing what the mixed layer depth should be.*

Please see the above plot in an answer to reviewer #1, where we have shown the relevant part of Fig.13 in Danabasoglu et al. (2014), with a reference and an ocean stand-alone simulation using the LR grid. We added the following text:

*"This is a feature that has already been identified in uncoupled FESOM simulations using the LR grid as part of the CORE-II intercomparison project (Danabasoglu et al, 2014; their Fig.13). Since the Gulf Stream in the LR (and REF) simulations is too zonal and reaches the northeastern North Atlantic, part of the flow has to downwell here, which we suspect could explain part of this deficiency by entraining waters and deepening the mixed layer.*
*Other factors influencing the mixed layer depth could be biased buoyancy fluxes or the restratification process via eddy activity."*

*o As you are talking about deep bias, I think is is worth adding discussion about the Nordic Sill overflow. Is the representation of the Nordic sill in your various configuration affect your conclusions?*

Although it could impact deep ocean biases as well, we have not analyzed the Nordic Sill overflow in this study. Similar to what we have written about the Mediterranean overflow, it would require much more systematic efforts. In this study, we used available ocean grids that were based on different mesh design principles (see section 2.2) and the representation of the Nordic Sill overflow was only one of many target quantities.

*o 4 supplementary documents in half a page of discussion I found it too much. Could you find a way to represent the point you want to make in a figure? Often reader like me do not take the time to get back on their browser, find the link, click on it and watch 4 movies.*

In order to keep a reasonable number of figures, we only added one new figure for illustrating the story about the Southern Ocean bias development, which was difficult to see in the movies (see Fig.11 below).

However, we agree that 4 supplementary documents can be rather overwhelming and we reduced the number of referenced supplementary movies to two in this subsection. Since the propagation of the bias along relevant isopycnals is already shown is Figure 8 (second and last row for LR and HR, respectively), the revised subsection now only refers to Fig. 8 (rather than to movies S3 and S4 as well). In the caption to Fig 8, the interested reader will still find the link to the animated versions.

*o You focus on the large improvement between LR and HR, but I found that there is also a large improvement between REF and LR (it let suggest also that the atmospheric model resolution is also important in decreasing the bias in depth.). See comments above on maybe removing REF from the document as LR and REF has roughly the same resolution.*
*o Please reformulate the conclusion of this section based on the comments above.*

In contrast to the improvements seen in the salinity profiles between REF and LR below 1500m, which we state now in the revised manuscript, we cannot identify a large improvement in potential temperature sections for years 91-100 between the LR and REF simulations (Figure 8). However, we added a sentence why we focus on LR and HR in this subsection:
*"To isolate the influence of the chosen ocean grid using the same atmospheric T127 configuration, we will focus on the LR and HR configurations here as examples."*

*o About the SST bias you mention at the end, please mention a reference to a figure.*

Done, we now refer to Fig.7b--d.

*- Displacement and tilt of isopycnal:*
*o You explain why the slope of the isopycnal is different but I think you should add clearly, why this leads to temperature bias along the isopycnals?*

Please refer to the answer to reviewer #1 above (hen-and-egg problem).

*o All your paper is focussing onto the depth 1000m. So I suggest for clarity to remove the discussion on the 200-300m depth range P15L7 to L14.*

Please see the answer to reviewer #1, where we rephrased this paragraph.

*o In your supplementary materials we clearly see in the LR case an error propagating from the Good Hope cap toward south America. Do you know why this propagation and not a bias intensifying all along the Atlantic Southern ocean?*

We don't think that the behaviour in the Southern Ocean is similar to what we wrote about the error propagation in the North Atlantic. In the Southern Ocean, although it looks like an error propagation in the supplementary material, it could rather be a bias that develops uniformly along the whole Atlantic Southern Ocean, depending on how eddies are treated and how the GM parameterisation is applied. We checked that the term "propagation" is just used for the North Atlantic case in the manuscript.

*o You mentioned that this strong bias in the Atlantic is due to difficulty of GM to balance the Ekman transport. So, why the error is so large in the Atlantic sector only? The other sector are quite good in LR and REF compare to HR.*

The Atlantic sector of the Southern Ocean is the place where the waters of North Atlantic origin upwell (governed by a balance between westerly winds and by eddies, which try to flatten the slope of isopycnals), and part of it is returned back to form the AMOC. One could say that this crucial process distinguishes the Atlantic sector from the Pacific sector, since the Pacific MOC does not show the same behaviour as known from the AMOC streamfunction. Biases in isopycnal slope in the Atlantic sector will be seen as temperature biases, as explained in an answer to reviewer #1 above.

*- Initialisation method:*
*o P15L17:I found the mention of 'usually based on a smoothed climatology as done in this study' confusing. I suggest to remove it. If you effectively smoothed the climatology, mention it in the previous sentenced and in the model configuration section.*

We did not smooth the climatology; we wanted to say that climatologies used for the initialization of ocean models are typically rather smooth. We changed "smoothed" to "rather smooth".

*o P15L19: what is the time scale you imply exactly by 'fast' adjustment? days? months? years?*

We imply a time scale of a couple of months, up to a year. We added:
*"After this first phase of fast adjustment, which takes months to one year …"*

*o About the example you mention (bias in the east North America), I will be more cautious. I agree that if the Gulf Stream is to north, you will have a warm bias in the Northern*

*Recirculation Gyre but based on the information you show, we don't know if PHC is representing this coastal area with strong boundary current correctly (you have strong temperature front in this area). You should at least put PHC sst in Figure 7.*

The strong boundary current and its temperature front is clearly seen in the PHC data set as plotted below. As suggested by the reviewer, we added this panel to Figure 7 and rewrote the caption accordingly.

[Figure]

*o Could you add precision about the time needed for the eddy fields to develop in your configuration?*

The time needed for the development is about 20-30 years, as we have seen in previous simulations with higher-frequency output. This fits the timescale cited in the manuscript (Allison et al., 2010).

*o P15L28: I think you should add a specific plot to show this instead of claiming it 'evident' on a supplementary material video. I had to watch back and forth frame by frame to be convince.*

As suggested by the reviewer, we added a new Figure 11 which shows the relevant frames of the supplementary movies.

[Figure]

**Fig.11:** *Southern Ocean maps of along-isopycnal potential temperature biases in **a)** LR and **b)** HR for years 1--10 and 31--40. Black contours show the outcropping location of the $\sigma_1$=31.8 isopycnal. For animations of the bias development with a 10-yr running window, see supplementary animations S1 and S2 (Rackow et al., 2018b).*

We now refer to the new figure in the main text (rather than to the movies) and the interested reader can still have a look at the animations (mentioned in the caption of Figure 11): *"Interestingly, in HR, an initial movement of the 31.8 isopycnal surface contour in the Southern Ocean towards the equator apparently leads to larger initial biases than in LR (Fig.11, left), and then it returns back to the south after 20 or 30 years. In years 31--40, the biases seem to recover and are again smaller than in LR (Fig.11, right)."*

*- Conclusions:*
*o In the model configuration section and introduction, you insist a lot on the local resolution, its benefice to run climate model. I was expecting it to be mentioned at the beginning and in a stronger way than you did.*

We agree and now mention this capability in the first paragraph of the Conclusions:
*"We found that the deep bias seen in AWI-CM-LR and REF is systematically reduced when moving to successively higher resolutions (10 km and higher) in eddy-active regions, a capability supported by FESOM1.4's use of multi-resolution ocean grids. Although there is certainly scope for improved eddy parameterizations, our results thus highlight the benefit of using high-resolution ocean components in climate modelling."*

*o I found the word 'the three worst performing CMIP5 model' not well chosen here without mentioning the criterion used for the assessment.*

This was certainly an unfortunate formulation without mentioning the specific criterion. The formulation has been changed to *"However, we could not identify a clear dependence of deep ocean biases on the vertical mixing schemes used in CMIP5 models: the three models with the strongest absolute error at a depth of 1000 m (GISS-E2-R, MPI-ESM-LR, GFDL-CM3; ...) use either KPP or PP mixing".*

*o Rewrite the discussion on Gibraltar based on the comments on the overflow and*

*Med. Sea water property.*

We extended the discussion stating that much more systematic efforts are still required to study the Mediterranean outflow, and that --besides the shown sensitivity to resolution-- an overflow parameterization might be a necessary next step to improve the model performance:

*"We identified two major sources for the deep ocean biases in the Atlantic ocean. The first source is the Strait of Gibraltar, which is likely to be a geometric issue related to the spatial discretization of this narrow strait (15 km) at relatively coarse resolution that is typical for CMIP5 models (about 100 km), and that often leads to increased Mediterranean outflow (e.g. Sterl et al., 2012). Much more systematic efforts are required to tune the horizontal and at the same time the vertical discretization of the Strait of Gibraltar. [...] We suspect that [...] the addition of an overflow parameterization might be necessary steps to further improve the model performance."*

*o P17L33: By 'outcropping often happens too far to the north compared to observations', please clearly specify what you imply? Do you imply that isopycnals outcrop in a region with stronger heat fluxes, warmer atmosphere ...?*

We want to point out that by changing the outcropping position to the North, denser water masses will be in contact with atmospheric conditions (fluxes) that are usually in contact with lighter waters. This shift will typically be accompanied by biases in (surface) temperature, for example, as mentioned earlier. We added:

*"Thus, outcropping often happens too far to the north compared to observations, so that denser water masses will be in contact with atmospheric conditions (fluxes) that are usually in contact with lighter waters, which can impact water mass transformation."*

*o Most of your paper is on the deep bias and you mentioned an example of bias developing at 200m depth. As I mentioned earlier, to keep your paper focus you should maybe get rid of the paragraph discussing this.*

Please see our earlier reply.

*o P18L5 to L13: You should move this paragraph earlier in the conclusion, maybe at the beginning.*

We thank the reviewer for this suggestion. Indeed, the conclusion reads more natural now that this is the second paragraph, right after we highlight the multi-resolution capability of AWI-CM (as also suggested by the reviewer).

*o P18L29: 'we have shown major improvement'. You need to add limitation to this statement. You only show major improvement on the T/S bias at 1000m. We don't know at all if it improve the MOC, MHT, bottom water formation.*

We changed the statement to:

*"Overall, we have shown major improvements when using medium-resolution (MR) and high-resolution (HR) meshes on representing the hydrography in the deep ocean around 1000m. These improvements at depth do not come at the expense of degradations in other climatically relevant fields, as shown by a performance index analysis (Appendix B)."*

**3 Figure and table comments**

*- Fig. 1:*
*o replace left/right by 'a)' and 'b)' and add it on the figure*
*o Comments on what you should see 'In the first hundreds meter ...' should go into the text not in the caption.*
*o Mean abs. error in the top 300m is hard to see (overwritten by blue and red line), maybe consider using transparency and envelope.*

We added labels to the figure and moved part of the comment on what can be seen to the text. By changing the order of plotting, we were able to increase the visibility of the mean absolute errors in the top 300m (black, plotted last).

*-Fig. 2: remove XR if you follow my comments on removing XR from the text.*

See comments above.

*-Fig. 3:*
*o As for Fig. 1, replace left/right by 'a)' and 'b)' and add it on the figure*
*o Comments on what you should see 'With the medium- and ...' should go into the text not in the caption.*

We added the labels to the figure and changed the caption accordingly. Although the comment is already given in the text, we'd like to repeat this statement here for readers who quickly scan papers, because it is the key message of the figure and a key message of the paper.

*-Fig. 6: split left column from the right column and put a label for each figure and use it in the caption.*

As suggested by the reviewer, we split the figure into a left and a right column and added the labels a) and b) to them.

*-Fig. 7: add a label for each figure and add PHC sst figure and maybe use the same colorbar as in figure 6.*

Done. We also added the SST plot for SST in Fig.7a. While a) and e) now show the SST and surface density structure in PHC, b)-d) still show the SST bias in the 3 simulations with respect to PHC and f)-h) show the surface density structures in the simulations. We kept the

colorbar as it was because the surface bias plots did not improve with the smaller range typical for the biases at depth in Figure 6.

*- Fig. 8: remove red line, as they are not commented on this figure.*

We still decided to keep the red lines in the figure for better orientation, and for consistency between figures 8 and 10.

*-Fig. 9: If you remove discussion on 30.8 and 30.5 isopycnal line, do not forgot to remove it here.*

We kept the lighter isopycnals (colored red) as mentioned in an answer above.

*You are not commented the green line in this figure, so please remove it.*

We refer to the green line in the manuscript: *"The mixed layer (green line and shading in Fig.8) is deep enough so that surface biases can reach the 31.8 and neighboring isopycnals, from where the signal is further advected towards the south. "*

*- Supplementary movies : please add a date on each frame, so we know where we are when we look at it (discussion on initial condition)*

Done. We added a time axis to Version 2 of the movies that are archived at Zenodo (https://doi.org/10.5281/zenodo.1323333). The DOI resolves to the latest version of the movies.

*- Table 1: Add interannual std and climate change signal in the top or bottom cells.*

We added the interannual std. dev. as well as climate change signals (RCP4.5 and 8.5) for the CMIP5 models in three new columns. We thank the reviewer for this suggestion because it revealed an interesting connection. The strength of the climate change signal appears to be somewhat linked to the quality of the ocean mean state (the magnitude of the error): models with smaller errors tend to simulate less pronounced climate change in the global ocean while models with larger errors in the mean state tend to simulate a stronger climate change in the ocean.

We added this preliminary finding to the introduction: *"[...] the magnitude of the projected climate change in the ocean appears to be ordered according to the models' mean absolute errors in the ocean (Table 1)."*

*- Table 2: Remove XR line and remove the internal name (not used in the manuscript).*

We'd like to keep the name because it guarantees better comparability with previous papers from our team, where names like "CORE" or "GLOB" were used. The renaming to "LR", "MR", "HR" and so on is only a relatively recent development in the course of our institute's

participation in CMIP6. We therefore changed "internal name" to "previous name" in the table. Why we wish to keep XR in the manuscript has been discussed in the above answers.

[revised manuscript text omitted]

---

## Author Response (AR2)

We thank both reviewers once more for their constructive and thorough comments. Our responses to the comments are given below. The reviewers' comments are in blue, our responses are in black.

***Anonymous Referee #1***

*Second review of "Sensitivity of deep ocean biases to horizontal resolution in prototype CMIP6 simulations with AWI-CM1.0" by Rackow and Co-authors*

*Although the authors addressed the majority of the comments and suggestions of both reviewers, I think there are still a few lingering items that should be addressed before publication. The first concerns consideration of other metrics to access the overall impacts of going to higher resolution. The authors indicate that they added performance indices (PIs) for some ocean fields. However, as far as I can tell from the explanation in Appendix B, this effort only concerns 3D temperature and salinity fields. In my request, I specifically asked for AMOC, and northward heat and salt transports from all the cases. Perhaps one should also add the Antarctic Circumpolar Circulation (ACC) transport at Drake Passage to this list. I would like to see spatial distributions of AMOC from all the cases as well as line plots of northward heat and salt transports. Plus a table of ACC transports. Based on Fig. 12, I am expecting that AMOC will be rather weak in MR.*

The performance indices are based on 3D temperature and salinity fields and were our attempt to evaluate the performance of the different ocean configurations in a comprehensive manner. It was not our intention to pass over the specific metrics asked for by the reviewer. These are now discussed in the following.

[Figure]

*Figure: The Atlantic Meridional Overturning Circulation (AMOC), in Sv (10^6 m^3/s), in the five pre-industrial experiments. REF/T63 shows the strongest AMOC, while the experiments with T127 atmosphere show a weaker AMOC. With increasing ocean resolution in the North Atlantic, the AMOC maximum appears to increase.*

As can be seen in the figure above, the AMOC in MR is not weaker and therefore does not stand out when compared to the other configurations. Generally, it appears that the change from T63 (REF) to T127 (all other configurations) reduces the AMOC strength significantly, which fits the earlier result by Sein et al. ("The Relative Influence of Atmospheric and Oceanic Model Resolution on the Circulation of the North Atlantic Ocean in a Coupled Climate Model", 2018). We added this reference to the paper.

[Figure]

*Figure: The meridional ocean heat transport (years 71-100) in the five pre-industrial experiments. The transport follows the known canonical picture of poleward heat transport.*

Regarding the northward ocean heat transport, it is rather similar between the different simulations. If any, the REF simulation with its T63 atmosphere stands out when compared to the other simulations.

We added a new Figure (3) with all AMOC patterns and the ocean heat transport to the manuscript as well as a short basic discussion in the new section 3.1 ("Atlantic Meridional Overturning Circulation (AMOC) and ocean heat transport").

[Figure]

*Figure: The meridional salt transport (years 71-100) in the five pre-industrial experiments. The transport given here corresponds to the term "Tr" in Treguier et al (2014, their Fig.4), which is about "-Tm".*

Depending on the details of the ocean model formulation (in particular the use of a linear free surface compared to the use of non-linear free surface), the "salt transport" will give different results. Linear free surface models, like the one used in this study, apply virtual salinity fluxes instead of surface freshwater fluxes while non-linear free surface models locally add/subtract mass to/from the ocean (depending on the sign of E-P-R, i.e. evaporation-precipitation-runoff), which results in a compensating barotropic ocean circulation response ("Tm"). This compensating response is small; but it is missing in all linear free surface models. The "salt transport" diagnosed in ocean models is therefore a rather ambiguous quantity since in equilibrium, the salt transport should be close to zero for any given latitude. Given the small differences between the simulations (only REF with T63 atmosphere stands out), we think this diagnostic should not be part of the revised document.

Reference:
*Treguier, A. M., et al. (2014): Meridional transport of salt in the global ocean from an eddy-resolving model, Ocean Sci., 10, 243-255,*
[https://www.ocean-sci.net/10/243/2014/os-10-243-2014.pdf](https://www.ocean-sci.net/10/243/2014/os-10-243-2014.pdf)

Regarding the transports of the Antarctic Circumpolar Current (ACC) at Drake Passage, the transports for our configurations are as follows:
REF: 153.7 Sv
LR: 213.1 Sv
MR0: 186.1 Sv
MR: 186.4 Sv
HR: 195.6 Sv

Again, the REF simulation with T63 somewhat stands out while the other configurations with T127 atmosphere are in closer agreement. REF is close to the observational estimate for the ACC transport, which is 136.7 ± 7.8 Sv (Cunningham et al., 2003). The mean ACC transport of current CMIP5 models is however 155 ± 51 Sv after Meijers et al. (2012), with range of 90 Sv up to 264 Sv. This means that all analysed configurations, including MR and HR, simulate an ACC transport within the typical model spread.

References:

*Cunningham et al. (2003): Transport and variability of the Antarctic Circumpolar Current in Drake Passage, JGR:Oceans, https://doi.org/10.1029/2001JC001147*

*Meijers et al. (2012): Representation of the Antarctic Circumpolar Current in the CMIP5 climate models and future changes under warming scenarios, JGR:Oceans, https://doi.org/10.1029/2012JC008412*

*Second, it is clear from Table 3 that the improvements are not really monotonic with increasing resolution. Specifically, going to HR and MR0 do not improve the solutions as far as the metrics in PI are concerned. So, statements like "We show that increasing ocean resolution locally to resolve ocean eddies leads to a major reduction in deep ocean biases." are not quite correct without a qualifier.*

We weakened the sentence in the abstract as follows:
*"We show that increasing ocean resolution locally to resolve ocean eddies leads to reductions in deep ocean biases, although these improvements are not strictly monotonic for the five different ocean grids."*
As discussed below, we also adjusted similar statements in other places of the text.

*A third item concerns the temperature and salinity biases depicted in Fig. 6 for the Strait of Gibraltar region. Specifically, I suspect that those biases are simply reflecting a vertical shift / change of the Mediterranean Outflow waters by a few hundred meters. Because these plots are at a constant depth of 1000 m, such shifts are not captured. I suggest adding vertical profiles of regionally-averaged potential temperature and salinity to capture such shifts. Although these items are not necessarily onerous, I would like to classify this revision as major because they can impact the conclusions*

The reviewer is right that the focus on a constant depth neglects possible vertical shifts. We thus added these further diagnostics to the new Fig. 8 (b and c), because of their probable

link to the spatial discretization of the Strait of Gibraltar:

[Figure]

*Figure: Vertical profiles of regionally-averaged potential temperature and salinity in the vicinity of the Strait of Gibraltar (5°W--30°W and 20°N--40°N). The horizontal dashed line highlights the depth of 1000m.*

It can be seen that REF/LR (and MR0) generate too much Mediterranean Outflow Water at 1000m depth, while MR and HR get not enough to a depth of 1000 meters. Besides the other mechanisms discussed in the paper, we think this diagnostic lends further weight to the idea that a systematic study of the width of the Strait (and of the representation of the adjacent bathymetry in the Gulf of Cadiz) is one possible way forward. There is untouched potential to better represent the local hydrography, because the simulated model profiles envelop the observed profile from PHC. We added this information to section 3.4.1.

*Minor items:*

*Fig. 1 and related discussion: Indicate how many CMIP5 models are included. Explain why DJF was chosen.*

Fig. 1 is based on the 13 CMIP5 models given in Table 1, where we state "CMIP5 models considered in the illustration of the deep ocean bias in Fig. 1". We now added this number explicitly to the caption of Fig.1 and to the main text:
*"This systematic error (Table 1) is illustrated by comparing temperature profiles from 13 CMIP5 historical runs (Fig.1b) with the PHC3 climatology".*

Our performance analysis with respect to CMIP5 models (described in Appendix B) is done for different seasons, and Fig.1 was therefore readily available for the different seasons. The figure is virtually identical when other seasons are considered, such as JJA (see plot below). We therefore chose to show DJF only:
*"As seasonal variability is low in 1000m depth, the bias is very similar for different seasons (not shown)."*

[Figure]

*Figure: Biases in CMIP5 models with respect to the PHC climatology [PHC 3.0, updated from: Steele et al. 2001]. a) Ensemble mean JJA potential temperature bias [K] at a depth of 1000m in 13 CMIP5 historical simulations for the period 1971---2000.  b) Individual depth-profiles of the mean absolute potential temperature error in the considered CMIP5 models (black lines).*

*Both here and elsewhere, change depth axis values to be positive because depth is positive downwards and negative values indicate a vertical coordinate that is positive upwards.*

We removed the minus from the depth axis in Fig.1. We also did this for Fig.4, 5, 6, 10 and 11.

*p.3, l.2: Please explain what "mapping" refers to.*

We use this rather mathematical or computational term here, because we think it nicely describes how the deep ocean is linked/connected to specific locations at the surface, via outcropping isopycnals. This is in analogy to how a function y=f(x) "maps" a value x (surface isopycnal contour) to another value y (e.g. ocean in 1000m), following the rule described by the function f (the sloping isopycnal surface in this case). The first occurence of the word "mapping" is on p.2 (l.8), and we have explained the term as follows:

*"This could lead to a wrong "mapping" of the deep ocean to the surface; in other words, this could link the deep ocean to incorrect locations at the surface, which may result in erroneous water mass formation."*

*p.3, l.33: Clearly state that the simulations are 100 years long.*

Done: *"In this study, we will analyze monthly-mean output of five 100yr-long pre-industrial simulations."*

*Table 1 caption: "…. at 1000 m depth."*

Done.

*p.4, l.11-12: This sentence is not clear. Please rephrase.*

We split the last sentence on p.4 and have rewritten it as follows:
*"For the Arctic, changing the thresholds can result in too diffuse boundary currents (Wang et al., 2014). Ultimately, these thresholds should be chosen automatically and separately for differently resolved regions of the global ocean. Their optimal choice thus remains an important research topic for multi-resolution climate applications."*

*Figure 6: Either use K or degree C throughout the manuscript. This figure uses degree C, elsewhere it is K.*

We changed the unit to K in Figure 6 (now 7; and for the other figure where differences were previously shown in °C, new Fig.11). It is now consistently used throughout the manuscript. For the absolute sea surface temperature (SST) field in Fig.7 (now Fig.9), however, we decided to leave the more common unit (°C). Therefore the difference to PHC is still shown here in °C as well to be consistent within the figure.

*p.14, l.11-12: This is true for a 30-year mean. What about seasonal and inter-annual excursions? If there are such excursions to deeper levels, then this general statement is not correct which can change your conclusions.*

This is a good point raised by the reviewer and we have weakened the statement accordingly. We explicitly added to the caption of the new Fig.9 and to the main text that -from the 30-yr means- we do not expect a major impact on our analysis:

*"... the strong cold temperature spot (...) is, however, not in direct contact with the deep ocean around 600--1000m depth via outcropping isopycnals (as diagnosed from 30-yr annual means). Despite possible seasonal excursions, we therefore do not expect a major impact on the analysis of the present study, which is focused on the deep ocean."*

*p.14, l.26-20: Can be deleted as already discussed earlier.*

We deleted *"...and higher resolutions are needed to get a more realistic outflow. Furthermore, an overflow parameterization or additional physics like tides could further improve the spreading of Mediterranean Waters into the North Atlantic."* as this was discussed earlier.
Based on the new T/S profile analysis suggested by the reviewer, where the climatological value from PHC at 1000m lies within the model estimates, we now also mention the untouched potential to get closer agreement between the model and the climatological profiles:

*"However, simply increasing the resolution in the Strait of Gibraltar does not automatically remove the bias; instead, climatological T/S profiles in the vicinity of Gibraltar lie between the according REF/LR/MR0 and MR/HR profiles (...). As mentioned before, a systematic geometric tuning of the ocean bathymetry in this area was not attempted (...), and there is thus potential for closer agreement with climatological potential temperature and salinity*

*profiles in this region by adjusting the spatial resolution within (and in the vicinity) of the strait."*

*Fig. 9 caption: Is there a missing URL?*

We updated the URL in the caption to a working link (last access 17 May 2019): http://www.soest.hawaii.edu/pwessel/gshhs/index.html

*p.18, last line: "…. climatology as the circulation and eddy fields are …."*

Done (p.17).

*Fig. 11 caption: These are not maps of along-isopycnal fields, but rather fields on a constant isopycnal.*

We have rewritten as *"Southern Ocean potential temperature biases [K] with respect to PHC, on the constant isopycnal $\sigma\_1=31.8$, in a) LR and b) HR for years 1--10 and 31--40."* (now Fig.12)

*p.20, l.4: "is systematically reduced when moving to successively higher resolution …." Not justified.*

We are more cautious in the revised manuscript and deleted "systematically" and "successively" from the statement. We also mention that the changes are not always strictly monotonic:
*"While the improvements are not strictly monotonic, we found that the deep bias seen in AWI-CM-LR and REF is generally reduced when moving to higher resolutions (10km and higher) in eddy-active regions, a capability supported by FESOM1.4's use of multi-resolution ocean grids."*

**Anonymous Referee #2**

*"Sensitivity of deep ocean biases to horizontal resolution in prototype CMIP6 simulations with AWI-CM1.0"*
*by Thomas Rackow, Dmitry Sein, Tido Semmler, Sergey Danilov, Nikolay Koldunov, Dmitry Sidorenko, Qiang Wang, and Thomas Jung*

*I am overall very pleased of the detailed replies given by the authors on my comments and the changes on the figures. I recommend a minor revision.*

We are very glad to hear that our reply answered the detailed comments by the reviewer. In the following we will address the remaining minor comments.

*Below my comments:*

*- Initialisation strategy section: As the authors did not add any results on it, I suggest it should be move out of the "results" section. It is more an idea than a results. It could fit in the conclusion as perspective.*

We agree and moved the section out of the "Results" section. We made a new independent section after the "Results" section, entitled "Perspective and implications for model initialization". We think this is a reasonable compromise since the Conclusions are already rather lengthy, they don't allow for subsections to structure the text, and it is uncommon to discuss a new Figure in the Conclusions section (Fig.11 [now 12] is introduced and discussed in the paragraph in question).

*- Vertical coordinate/resolution: In order to ease a bit the reading, may I suggest to replace "there are in total 46 levels (at 0, 10 .... 5650 and 5900m)" by something like there are in total 46 levels with a resolution varying from 10 m in surface to 250 m deeper than 2150 m.*

We added a better readable text as follows:
*"... there are in total 46 levels with vertical resolution ranging from 10m at the surface to 250m below 2150m."*

*- on the bathymetry: You include a web link in the article, could you check the policy on this, I think it should go on footnote or biblio with precision on when you access it the last time. I am surprised that there is no technical note or reference for the GEBCO 1 min data set.*

We checked the GMD author instructions at
[https://www.geoscientific-model-development.net/for_authors/manuscript_preparation.html](https://www.geoscientific-model-development.net/for_authors/manuscript_preparation.html)
and therefore added a new entry in the references, with last access date:
*"GEBCO: One Minute Grid, last access: 4 April 2019,*
*https://www.gebco.net/data_and_products/gridded_bathymetry_data/gebco_one_minute_grid/, 2008."*

*- Figure 11 : What are the white area ? Is it the sea ice area ? If yes, mention it. But if possible to keep consistency with figure 10, you should not print the sea-ice area. Please also mention in the caption that the bias in with respect to PHC.*

Fig.11 (now 12) shows biases on an isopycnal, and black contours show the outcropping location of the isopycnal. To the south of the black contour, there is therefore no data to be shown (=white) because the isopycnal already reached the surface.
In contrast, Fig.10 (now 11) is different in the sense that it shows a globally defined field (SST) where the outcropping location of an isopycnal is just overlaid.

We rewrote in the caption as follows: *"Southern Ocean maps of along-isopycnal potential temperature biases [K] with respect to PHC in **a)** LR and **b)** HR for years 1--10 and 31--40."* and *"Areas to the south of the outcropping location are white (indicating no data)."*

*- Geothermal flux precision: this sentence should go in the model description.*

We moved the sentence to Section 2, "Model configuration".

[revised manuscript text omitted]

---

## Author Response (AR3)

The editor's comments are in *italics*, our replies are in blue.

*Dear Author,*

*Thank you for your careful review of your manuscript following the 2nd round of reviews. I think that your manuscript is now almost ready for publication but I would like you to consider the following comments from my side for its final version:*

Thank you very much for the positive evaluation of our paper. We are glad that our response answered most of the questions and have considered the four remaining comments in the final version of the paper. In addition to these changes, we want to note that we added a data-DOI to the sentence *"This paper does not document AWI's final CMIP6 pre-industrial control simulations (Semmler et al., 2018) ... "* in the last paragraph of the Conclusions section, because the final CMIP6 data will become available at the provided link. Moreover, a second institution was added for one of the co-authors.

In the following, please find a point-by-point answer to the four editor comments.

1- *Regarding the first comment of referee #1, thank you for including the AMOC and ocean heat transport figures and analysis. Your arguments on why not to include the meridional salt transport are convincing to me. However, I think you should answer the referee's demand to include ACC numbers and analysis, as you do in your reply (but not in the revised manuscript). Thank you for doing so in the final version of the manuscript.*

Of course, we added a new table with the ACC transports from the AWI-CM configurations and CMIP5 (Meijers et al., 2012), along with an observational estimate (Cunningham et al., 2003). We also added the discussion from our reply to the manuscript. The section 3.1 was renamed to "Atlantic Meridional Overturning Circulation (AMOC) and ocean transports" because both northward ocean heat transport and ACC transport are discussed.

2- *Regarding your coupling procedure, I am not sure why you include reference to (Valcke, 2013; Valcke et al., 2013 ) when discussing the bicubic mapping. Also, it is just when reading the code availability section that one can understand that you use OASIS3-MCT. It would probably be more appropriate to remove references to (Valcke, 2013; Valcke et al., 2013 ) but mention in section 2 that you are using OASIS3-MCT and make a reference to*

A. *Craig, S. Valcke, L. Coquart, 2017: Development and performance of a new version of the OASIS coupler, OASIS3-MCT_3.0, Geosci. Model Dev., 10, 3297-3308, https://doi.org/10.5194/gmd-10-3297-2017, 2017.*

Concerning the bicubic mapping, we used a formulation in the manuscript that is very close to the comment on p.378 in Valcke (2013), so this reference should probably be kept. However, we removed the reference to Valcke et al. 2013 and added the newer OASIS3-MCT reference, as suggested.

3- *Still regarding the coupling procedure, I don't understand what you mean by "direct coupling" and why this "…. effectively reduces the number of necessary*

*interpolations...";OASIS3-MCT still has to express the coupling fields of the source grid on the target grid, as the source and target grids do not match, right? Can you clarify (or remove this mention to the exchange grid and to this reduction of necessary interpolations, if you think it is complex to explain and not very relevant here)?*

Previously, FESOM -in coupled mode- used to interpolate its surface fields to an intermediate (regular) exchange grid internally (first interpolation step); from the coupler's perspective, the model thus looked like a model with a regular grid despite using unstructured triangular grids. The regular surface fields were then interpolated once more to the atmospheric grid by the coupler. In the new setup, the intermediate exchange grid is gone since OASIS3-MCT supports unstructured surface grids "directly", but you are correct that the source and target grids still do not match, so the coupler still has to express the fields on the target grid using interpolation. Since this is very technical information and not very relevant, we removed the entire sentence from the text as suggested.

*4- P.16, l.21 "the latter source", maybe clarify that you are talking about the north-eastern North Atlantic source, as the mention is somewhat far in the preceding paragraph.*

True, we have changed the sentence accordingly.

[revised manuscript text omitted]